# In vivo autofluorescence lifetime imaging of spatial metabolic heterogeneities and learning-induced changes in the *Drosophila* mushroom body

**Philémon Roussel[1], Mingyi Zhou[2], Chiara Stringari[3], Thomas Preat[2], Pierre-Yves Plaçais[2]\*, Auguste Genovesio[1]\***

[1]Computational Bioimaging and Bioinformatics, IBENS, École normale supérieure, CNRS, INSERM, Université PSL, Paris, France; [2]Energy & Memory, Brain Plasticity Unit, ESPCI, CNRS, Université PSL, Paris, France; [3]Laboratory for Optics and Biosciences, École Polytechnique, CNRS, INSERM, Institut Polytechnique de Paris, Palaiseau, France

**\*For correspondence:**
pierre-yves.placais@espci.fr (P-YP);
auguste.genovesio@ens.psl.eu (AG)

**Competing interest:** The authors declare that no competing interests exist.

## eLife Assessment

This **valuable** study uses NAD(P)H fluorescence lifetime imaging (FLIM) to map metabolic states in the Drosophila brain. The authors reveal subtype-specific metabolic profiles in Kenyon cells and report learning-related changes, supported by **solid** evidence and careful methodology. However, the FLIM shifts observed after memory formation in α/β neurons are small and only weakly significant, so the ability of FLIM to detect subtle physiological changes still requires further validation. Nevertheless, this work provides a strong starting point and demonstrates the promising potential of FLIM for probing neural metabolism in vivo.

**Abstract** Neuronal energy regulation is increasingly recognized as a critical factor underlying brain functions and their pathological alterations, yet the metabolic dynamics that accompany cognitive processes remain poorly understood. As a label-free and minimally invasive technique, fluorescence lifetime imaging (FLIM) of coenzymes NADH and NADPH (collectively referred to as NAD(P)H) offers the possibility to resolve cellular metabolic profiles with high spatial precision. However, NAD(P)H FLIM's capacity to detect subtle variations in neuronal metabolism has not been demonstrated. In this study, we applied NAD(P)H FLIM to map the metabolic profiles of *Drosophila* neurons in vivo across multiple scales, focusing on the primary centers for associative memory: the mushroom bodies (MBs). At a broad scale, we obtained an overview of the metabolic signatures of the main brain tissue and identified a marked difference between neuropil and cortex areas. At a finer scale, our findings revealed notable heterogeneity in the basal metabolic profiles of distinct MB neuron subtypes. Measurements performed after associative olfactory learning also uncovered a low-magnitude subtype-specific metabolic shift associated with memory formation, suggesting the utility of NAD(P)H FLIM in detecting physiology-driven changes linked to brain function. These results establish a promising framework for studying the spatial heterogeneities and the dynamics of cerebral energy metabolism in vivo.

## Introduction

Understanding the intricate relationship between brain function and energy metabolism is a fundamental challenge in neuroscience. Cellular metabolism relies on extremely conserved biochemical processes that sequentially catabolize primary energy substrates into usable energy in the form of

ATP. These ubiquitous processes are especially active in the brain, which consumes large amounts of energy relative to its weight (*Rolfe and Brown, 1997*), mostly in the form of glucose (*Sokoloff, 1999*). Neurons indeed have high and dynamic energy needs, due in particular to the maintenance of ion gradients and the propagation of action potentials (*Attwell and Laughlin, 2001*).

In vertebrates, astrocytes are major players in the achievement of this spatiotemporally fluctuating energy allocation. Astrocytes orchestrate the neurovascular coupling (*Bélanger et al., 2011*) but also store energy-carrying molecules and fuel them to neurons, thanks to their complementary metabolic profile. While neurons predominantly generate ATP through oxidative phosphorylation, astrocytes display a high level of glycolysis (*Itoh et al., 2003*; *Boumezbeur et al., 2010*), resulting, for a significant fraction, in lactate production (*Serres et al., 2005*; *Bouzier-Sore et al., 2006*). Numerous studies tend to demonstrate the net transfer of energy from astrocytes to neurons through a so-called lactate shuttle (*Pellerin and Magistretti, 1994*; *Pellerin and Magistretti, 2012*; *Bélanger et al., 2011*). Despite some differences, accumulating evidence shows that insect glia shares many properties with its vertebrate counterparts (*Nagai et al., 2021*; *De Backer and Grunwald Kadow, 2022*), including their highly glycolytic profile and their ability to transfer glycolysis-derived metabolites to neurons (*Volkenhoff et al., 2015*; *Rittschof and Schirmeier, 2018*).

In addition to the necessity of adapting energy supply to meet neuronal consumption, studies in diverse species, including insects (*Chandrasekaran et al., 2015*; *Plaçais et al., 2017*) and rodents (*Shimizu et al., 2007*; *Parsons and Hirasawa, 2010*; *Suzuki et al., 2011*), have shown that energy fluxes modulate higher brain functions. These findings support a shift away from a purely neurocentric perspective, suggesting that energy is not only delivered to neurons on demand but is instead regulated by an integrated neuron-glia network, actively shaping cerebral activity. Memory formation, in particular, has already been shown to be closely associated with neuronal energy regulation. Several studies have, for example, reported the importance of astrocytic metabolism during memory formation in vertebrates (*Gibbs et al., 2006*; *Newman et al., 2011*; *Suzuki et al., 2011*; *Gao et al., 2016*).

With its well-characterized genetics, neuroanatomy, and memory capacities, *Drosophila* serves as an ideal model organism for investigating the neural basis of memory. Aversive olfactory memory can be induced in *Drosophila* through classical Pavlovian conditioning, by pairing an odor with electric shocks (*Tully and Quinn, 1985*; *Modi et al., 2020*). The neurons encoding these memories are located in the mushroom bodies (MBs), which are considered the main brain center for associative memory in insects (*Heisenberg, 2003*; *Modi et al., 2020*). The *Drosophila* MBs are paired structures, including around 2000 intrinsic neurons, named Kenyon cells (KCs), in each hemisphere (*Aso et al., 2009*; *Aso et al., 2014*; *Li et al., 2020*). These neurons receive dendritic inputs from the antennal lobes in the MB calyces (*Keller and Vosshall, 2003*), located in the posterior part of the brain. Their axons form a bundle known as the peduncle, which extends to the anterior part of the brain before branching into horizontal and vertical lobes. KCs are categorized into three main subtypes based on their axonal morphology and genetic expression profile: α/β, α′/β′, and γ neurons (*Crittenden et al., 1998*; *Lee et al., 1999*). All three populations are considered to be involved at multiple stages of memory processing (*Krashes et al., 2007*), but converging evidence suggests that γ KCs are required for encoding and retrieving immediate memory (*Qin et al., 2012*; *Bouzaiane et al., 2015*; *Owald et al., 2015*; *Perisse et al., 2016*), while α/β neurons underlie a middle-term memory that develops more slowly over several hours (*Blum et al., 2009*; *Séjourné et al., 2011*; *Bouzaiane et al., 2015*).

Recent studies have underscored the importance of energy regulation in the MBs during memory formation. Increased mitochondrial energy flux in MB neurons (*Plaçais et al., 2017*) with precise spatiotemporal dynamics (*Comyn et al., 2024*) and fueled by the transfer of glycolysis-derived alanine from cortex glia or astrocytes to neurons (*Rabah et al., 2023*) indeed plays critical roles in the formation of middle-term and long-term memory. These findings position the *Drosophila* MBs as a compelling model for exploring how neuronal metabolism and cognitive processes intersect. In these works, metabolic dynamics in the MBs have been tracked using genetically encoded fluorescent sensors tracking specific metabolites.

The growing recognition of the role played by energy metabolism in brain function highlights the need for universal methods to monitor cellular metabolic profiles in vivo across species, with high spatial and temporal resolution. Cellular ATP production through catabolic pathways relies on redox reactions involving carbon-based energy substrates and redox couples, notably the coenzyme nicotinamide adenine dinucleotide (NAD), in its oxidized ($NAD^+$) and reduced (NADH) forms. During

glycolysis and the tricarboxylic acid cycle, $NAD^+$ is reduced to NADH, while NADH is oxidized back to $NAD^+$ in the mitochondrial respiratory chain and through lactic acid fermentation. Since NADH is autofluorescent and $NAD^+$ is not, NADH fluorescence can serve as a proxy for the cellular redox state. In practice, NADH fluorescence is difficult to isolate from that of nicotinamide adenine dinucleotide phosphate (NADPH), a coenzyme involved in a wide range of anabolic processes. Nevertheless, shifts in the combined fluorescence signal of NADH and NADPH – referred to as NAD(P)H – are generally thought to be dominated by the influence of the NADH in the brain (*Kasischke et al., 2004*). Fluorescence lifetime imaging (FLIM) enables differentiation between free and protein-bound NAD(P)H molecules based on their distinct fluorescence lifetimes. Free NAD(P)H exhibits a shorter lifetime – around 0.4 ns – while bound NAD(P)H has a longer lifetime, ranging from 1 to 5 ns (*Lakowicz et al., 1992*; *Kolenc and Quinn, 2019*; *Datta et al., 2020*). By analyzing the contributions of these distinct lifetimes to the overall fluorescence decay, FLIM provides a quantitative estimate of the relative concentrations of free and bound NAD(P)H, robust to intensity distortion caused by light scattering. The free-to-bound NAD(P)H ratio measured by FLIM has been linked to changes in the balance between glycolysis and oxidative phosphorylation (OXPHOS), with a higher proportion of free NAD(P)H indicating an increased reliance on glycolysis relative to OXPHOS, and vice versa. This relationship has been observed in vitro and ex vivo in various contexts, including metabolic perturbations (*Zheng et al., 2010*; *Stringari et al., 2012b*; *Drozdowicz-Tomsia et al., 2014*; *Liu et al., 2018*; *Yang et al., 2021*), cell differentiation (*Guo et al., 2008*; *Stringari et al., 2012a*), and carcinogenesis (*Bird et al., 2005*; *Skala et al., 2007*; *Rück et al., 2014*), underscoring the utility of FLIM in assessing cellular metabolic states.

In the present study, we propose to demonstrate that NAD(P)H FLIM can capture the metabolic heterogeneities and shifts underpinning memory formation in the *Drosophila* brain. In contrast to hypothesis-driven approaches that rely on metabolite-specific sensors, NAD(P)H autofluorescence provides a global readout of cellular metabolism. Furthermore, autofluorescence imaging can be applied in non-genetic experimental models where genetically encoded indicators cannot be introduced. As a label-free and minimally invasive technique, NAD(P)H FLIM holds great promise for profiling neuronal metabolism in vivo. Despite this potential, its application to the brain has been so far limited. It has been used in vivo in the rodent brain to monitor strong and widespread metabolic responses to anoxia (*Yaseen et al., 2013*) and pharmacological interventions (*Yaseen et al., 2017*; *Gómez et al., 2018*). However, its capacity to detect more subtle metabolic events associated with cognitive processes remains to be established.

Here, we employed FLIM of NAD(P)H to map the in vivo metabolic profiles of *Drosophila* neurons across multiple scales – from a broad view of the central brain to individual KC subtypes. This approach allowed us to visualize the cerebral metabolic heterogeneity at different levels and to identify a memory-related metabolic shift within a specific KC subtype. Our findings contribute to the understanding of the fine-scale energy regulation taking place in the MBs and establish NAD(P)H FLIM as a promising framework for future investigations into the energy dynamics underlying cognition.

## Results

### Metabolic landscape of the central brain highlights tissue-specific signatures

For this novel application of FLIM, we began by mapping the metabolic landscape across the entire *Drosophila* central brain, with a particular focus on the MBs. We conducted in vivo two-photon imaging of NAD(P)H FLIM over the entire central brain in flies expressing a DsRed cytosolic marker in MB neurons (MB-DsRed; Dataset 1 in Methods). Multi-exponential fitting and phasor analysis constitute the main approaches for NAD(P)H FLIM decay characterization (*Digman et al., 2008*; *Stringari et al., 2011*; *Kolenc and Quinn, 2019*; *Schaefer et al., 2019*; *Datta et al., 2020*). The former was chosen to automatically extract quantitative metabolic indicators. Given the novelty of the application, we opted for a simple and well-grounded approach to modeling NAD(P)H decays: the bi-exponential model (*Kolenc and Quinn, 2019*; *Schaefer et al., 2019*). The decay curves were spatially binned to enhance the signal-to-noise ratio (*Figure 1a*). A bi-exponential model was fitted to each decay using maximum-likelihood estimation (*Figure 1—figure supplement 1*), generating local estimates of the free NAD(P)

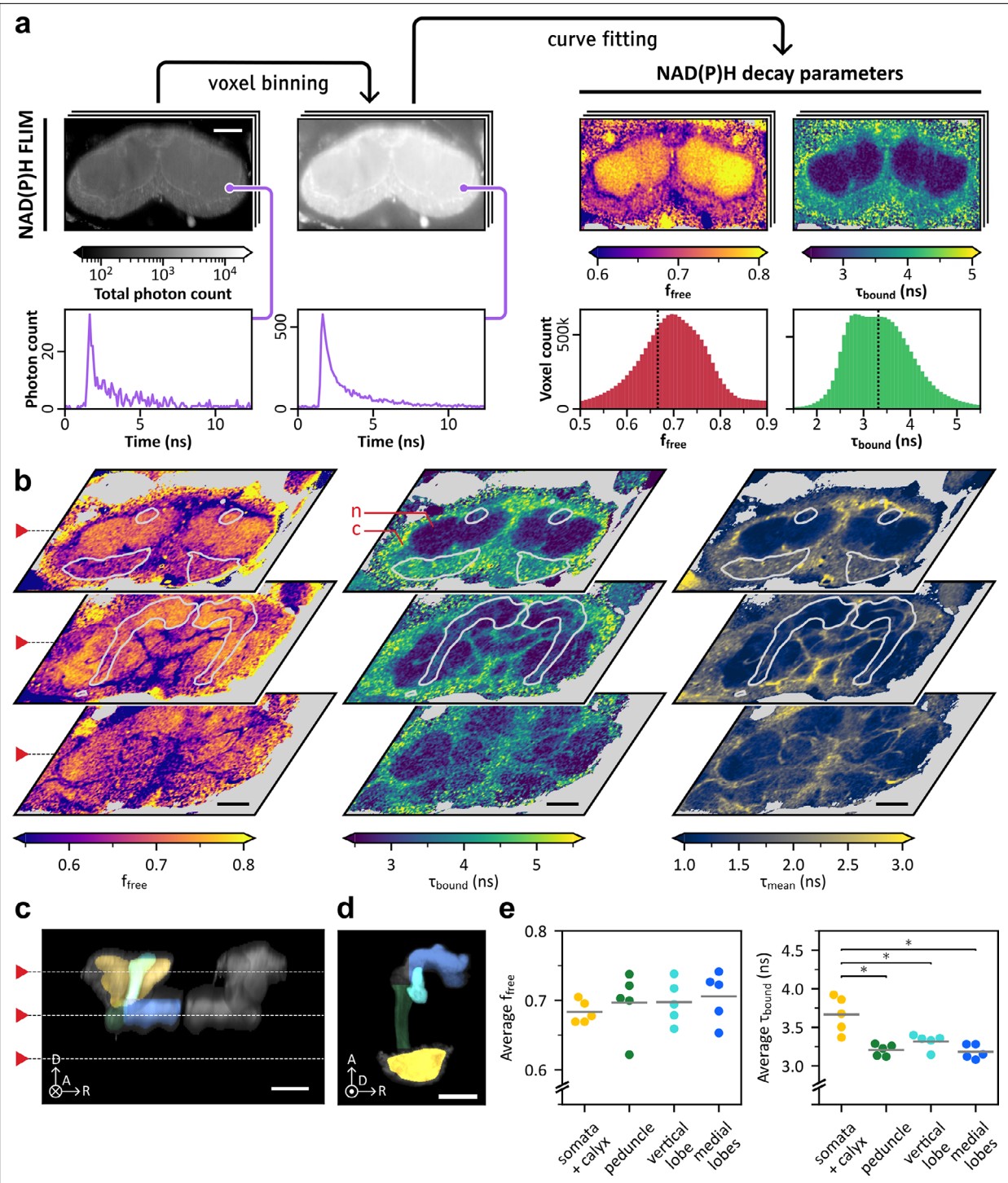

**Figure 1.** In vivo measurements of NAD(P)H state over the central brain and across mushroom body (MB) regions. Scale bars: 50 μm. Axes indicate anterior (A), dorsal (D), and right (R) anatomical orientations. The circled dot and cross indicate the directions pointing toward and away from the viewer, respectively. (**a**) Illustration of the main steps of the mapping of NAD(P)H decay parameters over the central brain. Fluorescence lifetime imaging (FLIM) images were spatially binned using a disk-shaped kernel (see Methods) and the resulting curves were fitted to extract decay parameters. The parameter histograms are derived from the whole 3D image. Each step is illustrated by a single horizontal section of the example 3D image. (**b**) Heatmaps of $f_{free}$, $\tau_{bound}$, and $\tau_{mean}$ over three horizontal slices of a single image. Contours of the MB, extracted from the MB-DsRed image, are superimposed in white. Voxels with photon counts under 500 are shown in gray. The cortex and neuropil are marked on the first slice of the $\tau_{bound}$ heatmap with red letters 'c' and 'n,' respectively. (**c**) Average of all transverse sections of the MB-DsRed image corresponding to panel b (gray), with colored segmentation regions superimposed on the left MB. White dashed lines denote the depth of the horizontal sections shown in panel b. (**d**) Horizontal view of a left MB

*Figure 1 continued on next page*

*Figure 1 continued*

showing the maximum intensity projection of the Kenyon cell (KC)-specific marker with superimposed segmentation regions. The circled dot indicates the direction pointing towards the viewer. (**e**) Within-subject averages of $f_{free}$ and $\tau_{bound}$ in the segmented MB regions, for five flies. Gray horizontal lines indicate the cross-subject mean values. No significant variations of $f_{free}$ were observed between the considered regions (repeated measures ANOVA, $F_{3,12} = 1.7$, $p=0.21$). Significant variations of $\tau_{bound}$ were observed between areas (repeated measures ANOVA, $F_{3,12} = 14.8$, $p=2.4\times10^{-4}$). Three post-hoc comparisons were performed to compare the somata and calyx region with the other areas. It showed that $\tau_{bound}$ was significantly higher in the somata and calyx (n=5, $\mu=3.67\pm0.23$ ns) compared to the peduncle (n=5, $\mu=3.21\pm0.07$ ns; paired samples Student's $t$-test, $t_4=4.7$, Bonferroni-adjusted $p=2.7\times10^{-2}$), the vertical lobe (n=5, $\mu=3.32\pm0.10$ ns; paired samples Student's $t$-test, $t_4=4.0$, Bonferroni-adjusted $p=4.8\times10^{-2}$) and the medial lobes (n=5, $\mu=3.18\pm0.09$ ns; paired samples Student's $t$-test, $t_4=4.3$, Bonferroni-adjusted $p=3.8\times10^{-2}$).

The online version of this article includes the following video and figure supplement(s) for figure 1:

**Figure supplement 1.** Examples of fit and distribution of 2I* for Dataset 1.

**Figure supplement 2.** Distributions of $f_{free}$ and $\tau_{bound}$ in different segmented mushroom body (MB) regions for the subjects of Dataset 1.

**Figure 1—video 1.** Spatial distributions of NAD(P)H properties over the volume of the central brain of subject 1.
https://elifesciences.org/articles/106040/figures#fig1video1

**Figure 1—video 2.** Spatial distributions of NAD(P)H properties over the volume of the central brain of subject 2.
https://elifesciences.org/articles/106040/figures#fig1video2

**Figure 1—video 3.** Spatial distributions of NAD(P)H properties over the volume of the central brain of subject 3.
https://elifesciences.org/articles/106040/figures#fig1video3

**Figure 1—video 4.** Spatial distributions of NAD(P)H properties over the volume of the central brain of subject 4.
https://elifesciences.org/articles/106040/figures#fig1video4

**Figure 1—video 5.** Spatial distributions of NAD(P)H properties over the volume of the central brain of subject 5.
https://elifesciences.org/articles/106040/figures#fig1video5

H fraction – $f_{free}$ – and the protein-bound NAD(P)H lifetime – $\tau_{bound}$ – across the central brain, for each fly. The weighted mean lifetime – $\tau_{mean}$ – was subsequently computed from these estimates.

The 3D spatial distributions of NAD(P)H properties could be visualized over the volume of the central brain (*Figure 1b* and *Figure 1—videos 1–5*). This mapping revealed the distinct metabolic fingerprints of large anatomical structures. In particular, the neuropil could be well distinguished from the cortex region, which contains neuronal somata and cortex glia, as it displayed higher $f_{free}$ and lower $\tau_{bound}$ values. Within the neuropil, layers with lower $f_{free}$ and higher $\tau_{bound}$ values stood out, delineating different regions, as visible on the $\tau_{mean}$ map. Based on qualitative visual evaluation, these structures likely correspond to the layers formed by processes of neuropil- and tract-ensheathing glia, as well as tracheal processes (*Pereanu et al., 2007*; *Kremer et al., 2017*).

To quantitatively compare the different MB regions, we coarsely segmented the labeled MB image into defined anatomical structures using thresholding of the MB-specific fluorescent red marker and manually defined separation planes. Due to the limited resolution of the cytosolic marker image at this scale, the cortical region of the MB, primarily consisting of KC somata enveloped by cortex glia, could not be isolated from the adjacent calyx, which contains KC neurites and axonal projections from external neurons. The resulting segmentation included the peduncle, the medial lobes and the vertical lobe, and the posterior part of the MB, consisting of the cortical region and calyx (*Figure 1d*).

The obtained segmentation enabled region-specific extraction of FLIM voxels, followed by the estimation of corresponding decay parameters. $f_{free}$ and $\tau_{bound}$ distributions were in most cases predominantly unimodal (*Figure 1—figure supplement 2*), allowing for effective representation by the mean. No significant variation in the average $f_{free}$ was observed across regions (*Figure 1e*, left). However, the posterior region showed notably higher $\tau_{bound}$ values than the rest of the MB (*Figure 1e*, right), consistent with what was observed at the whole brain level.

Overall, these results highlight regional differences in $f_{free}$ and $\tau_{bound}$ among the neuropil, the cortex region, and ensheathing glia in the central brain. At the MB level, a difference in $\tau_{bound}$ was still observed between the posterior part of the MB and purely synaptic regions. The interpretation of $f_{free}$ as an indicator of the glycolysis-to-OXPHOS ratio has theoretical support (*Schaefer et al., 2019*) and has been validated in various contexts (see Introduction). The interpretation of $\tau_{bound}$ is more complex, as it is generally thought to represent a weighted average of fluorescence lifetimes of different enzyme-bound NAD(P)H species and to be influenced by changes in the composition of the proteins binding to NAD(P)H molecules (*Kasischke et al., 2004*; *Yaseen et al., 2013*; *Yaseen et al.,*

2017; *Evers et al., 2018*; *Sharick et al., 2018*). Some studies also suggested that $\tau_{bound}$ variations may more sensitively reflect changes in the NADPH/NADH ratio than in NAD(P)H binding state, with $\tau_{bound}$ being positively correlated to the NADPH/NADH ratio (*Blacker et al., 2014*; *Blacker and Duchen, 2016*). Here, the difference between neuropil and cortical regions seems to be more strongly reflected by $\tau_{bound}$ than $f_{free}$, as confirmed by the quantitative analysis in the MBs, suggesting differences between these tissues in the composition of the proteins binding to NAD(P)H and/or in the NADPH/NADH ratio.

## Somata and calyx regions have different metabolic profiles

The neuropil region of the MB – comprising the calyx, peduncle, and lobes – contains KC neurites and is innervated by numerous additional neuronal types, including projection neurons, dopaminergic neurons, and MB output neurons as well as broadly-innervating DPM and APL neurons, which complicates the attribution of observed metabolic signals to particular cells. Isolating the NAD(P)H signal from the MB cortical region thus allows for the specific characterization of KC somata's metabolic profile and its comparison with that of the adjacent synaptic region.

To this end, we acquired images with a more restrained scope, centered on the cortical area of the MBs. Each acquisition captured the KC-specific cytosolic marker MB-DsRed and NAD(P)H FLIM in a single hemisphere (Dataset 2 in Methods). We analyzed these images using mapping tools derived from other datasets: a MB template and two segmentation masks, designed to isolate the KC somata and the calyx, respectively. The template image, designed to represent the average shape of the posterior MB region (*Avants et al., 2010*), was derived from a set of 165 preprocessed MB-DsRed images (*Figure 2a*; Dataset 3 in Methods) using a 3D volume registration procedure consisting of affine and non linear diffeomorphic transformations (*Figure 2b*; *Avants et al., 2008*). A common template was established for left and right hemisphere MBs (*Figure 2cd*) after confirming that their average shape differences were minimal (Supplementary results in Appendix 1). The masks were derived from MB-DsRed images and KC-specific nuclear marker images (*Figure 2—figure supplement 1*; Datasets 3 and 4 in Methods), using the same registration method. Once registered to each image, the masks enabled us to select the regions of interest in the FLIM image (*Figure 3a*). Decay parameters were then extracted as previously.

Registration was further used to project decay parameter images to the template space, enabling the generation of an average map of $f_{free}$ and $\tau_{bound}$ in the cortical region and calyx (*Figure 3b*) while minimizing the impact of inter-individual anatomical variations. This map was obtained by pooling left and mirrored right hemisphere images, after confirming that there was no hemisphere-specific spatial distribution of $f_{free}$ (Supplementary results in Appendix 1). This average map highlighted that $f_{free}$ was consistently higher in the calyx neuropil than in the cortical region of the MB.

To quantitatively substantiate this observation, the within-subject average was computed using both hemisphere images when available, or a single hemisphere image otherwise. The results confirmed that the calyx exhibits significantly higher values of $f_{free}$ than the KC somata region, with means of 0.76 and 0.69, respectively (*Figure 3b*, bottom right). According to the literature, the difference in $f_{free}$ likely reflects a higher reliance on glycolysis relative to OXPHOS in the calyx. Similar analysis of $\tau_{bound}$ revealed a significantly shorter bound-NAD(P)H lifetime in the calyx compared to the somata region (*Figure 3—figure supplement 1a and b*). This difference in $\tau_{bound}$ is consistent with the previously observed trends between the posterior region and the rest of the MB and, at a larger scale, between cortex and neuropil in general. Likewise, it could reflect differences in protein composition and/or in the NADPH/NADH ratio.

Measures of $f_{free}$ and $\tau_{bound}$ were notably anticorrelated in the posterior part of the MB (*Appendix 1—figure 3*). The anticorrelation between these parameters can be qualitatively observed in several studies (*Bird et al., 2005*; *Skala et al., 2007*; *Wetzker and Reinhardt, 2019*; *Yang et al., 2021*). Multi-exponential fitting has been reported to potentially introduce artificial correlations between parameter estimates (*Grinvald and Steinberg, 1974*; *Johnson, 2000*). Through simulation, we confirmed that the relationship observed in our data was not an artifact of the fitting process (Supplementary results in Appendix 1), supporting the hypothesis of its biological origin.

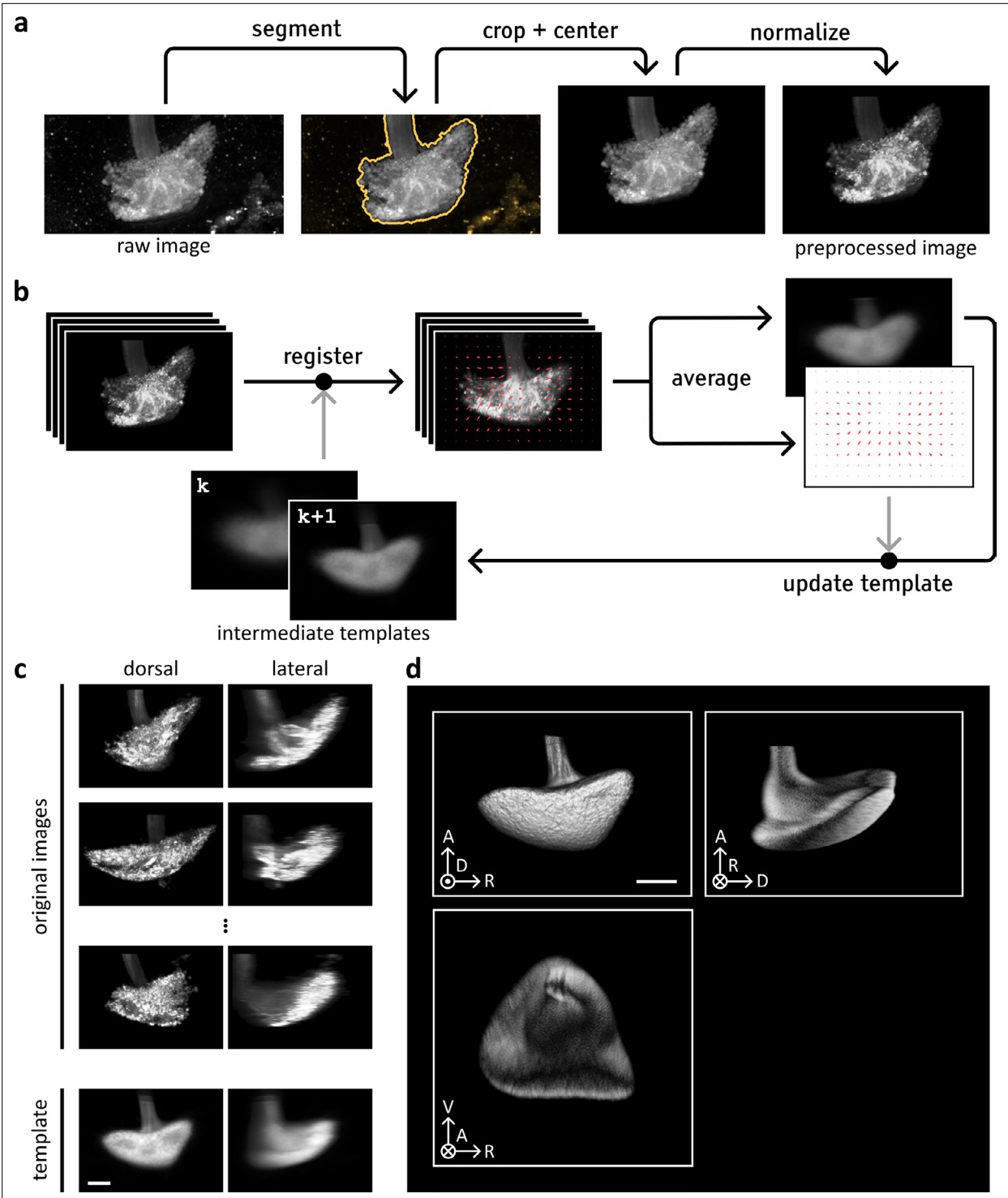

**Figure 2.** Establishment of a reproducible mushroom body (MB) template. (**a**) Illustration of the main steps of the preprocessing pipeline. This process was applied to the 165 MB images obtained from flies expressing fluorophore DsRed in the cytoplasm of all Kenyon cells (KCs) (Dataset 3). (**b**) Illustration of the template building process. An initial template is obtained by averaging all preprocessed images. The images are then registered to this intermediate template. The resulting registered images and transformations are averaged. A transformation proportional to the inverse of the mean transformation is applied to the average registered image to get an updated template image. The cycle was repeated several times to obtain the final template. (**c**) Examples of preprocessed MB images and final template. The left and right columns show dorsal and lateral maximum intensity projections of each image. (**d**) Views of the MB template image generated through volume rendering with ParaView software (*Ahrens et al., 2005*).

*Figure 2 continued on next page*

*Figure 2 continued*

Axes indicate anterior (A), dorsal (D), ventral (V), and right (R) anatomical orientations. View-aligned axes orientations depicted with dot (towards viewer) and cross (away from viewer). Scale bar: 25 μm.

The online version of this article includes the following figure supplement(s) for figure 2:

**Figure supplement 1.** Establishment of masks for the somata and calyx regions.

## Memory formation does not induce a significant metabolic shift over the MB somata and calyx regions

After imaging basal metabolism in the posterior MB region, we investigated whether memory formation induced detectable metabolic changes. To this end, we acquired images of the MB posterior region in conditioned flies (refer to Dataset 5 in Methods). Flies were subjected to a classical olfactory aversive learning protocol (*Tully and Quinn, 1985*). In this paradigm, flies are first exposed to an odorant (odor A) paired with a train of electric footshocks, and then to a second odorant without shocks (odor B; *Figure 3c*, top). Following a single training session, it is known that flies tend to develop short- (STM) and middle-term memory (MTM), expressed as learned avoidance of odor A compared to odor B (*Margulies et al., 2005*). Avoidance behavior is thought to result from a learned bias in the odor choice probability and is measured at the group level, through the comparison with control flies. Control flies undergo unpaired conditioning, consisting of an exposure to the aversive stimulus and then to both odors, sequentially. No significant difference in the within-subject averages of $f_{free}$ were observed in the somata or calyx regions between conditioned and control flies (*Figure 3c*, bottom). Similarly, no memory-associated shifts in $\tau_{bound}$ were detected in either region (*Figure 3—figure supplement 1c*).

In this experiment, indicators derived from NAD(P)H FLIM did not reveal a significant metabolic shift affecting globally the KC somata region or the calyx after a single training cycle. This result suggests the absence of an overall metabolic shift in these regions or its low amplitude relative to NAD(P)H FLIM sensitivity. Alternatively, specific effects may be diluted or averaged out, as the analyzed areas encompass diverse cell types that could exhibit distinct memory-related metabolic dynamics.

## KC subtypes have different basal metabolic profiles

To further explore the metabolic heterogeneity of the MB, we aimed at characterizing the basal metabolism of the main subtypes of KCs (α/β, α'/β', and γ). These subpopulations are known to have distinct functional roles. In particular, α/β neurons are specifically involved in middle- and long-term memories, whose formation has been shown to critically depend on energy regulation, while γ neurons are more specifically involved in short-term memory.

We initially attempted to separate subtype contributions within images through a mapping approach. To this end, we computed the average spatial distribution of each of these subtypes within the somata region of the template image. While anatomical maps of the MB neuropil have shown that the neurites of different KC subtypes form distinct bundles (*Aso et al., 2014*; *Li et al., 2020*), the stereotypy of KC somata distribution remained largely unexplored. Our analysis (Supplementary results in Appendix 1) revealed that these distributions were spatially non-uniform and reproducible from one group of flies to another (*Appendix 1—figure 5*). Comparisons of pairs of images from identical or different hemispheres revealed an absence of lateralization of the spatial distribution of the neuronal subtypes (*Appendix 1—figure 6*). However, the degree of overlap between the average subtype distributions was high (*Appendix 1—figure 7*), suggesting important inter-individual variations and making effective separation through segmentation masks impossible.

We, therefore, performed recording in flies expressing a nuclear marker in either α/β, α'/β', or γ neurons, to separately measure the metabolic profile of each subpopulation while minimizing the contribution of the surrounding tissues. For each fly, standard two-photon imaging of this marker was performed alongside NAD(P)H FLIM over the somata region of the MB, one hemisphere at a time (Dataset 6 in Methods). Nuclear marker images were used to isolate the fluorescence decays around the somata of the marked neuronal subtype (*Figure 4a*). Within-subject averages of $f_{free}$ were then computed from the masked FLIM images. These measurements revealed that the KC subtypes exhibit different basal metabolic profiles. Indeed, α/β, α'/β', and γ somata displayed decreasing $f_{free}$ values of 0.73, 0.71, and 0.70, respectively, with statistical testing showing a significant difference between α/β

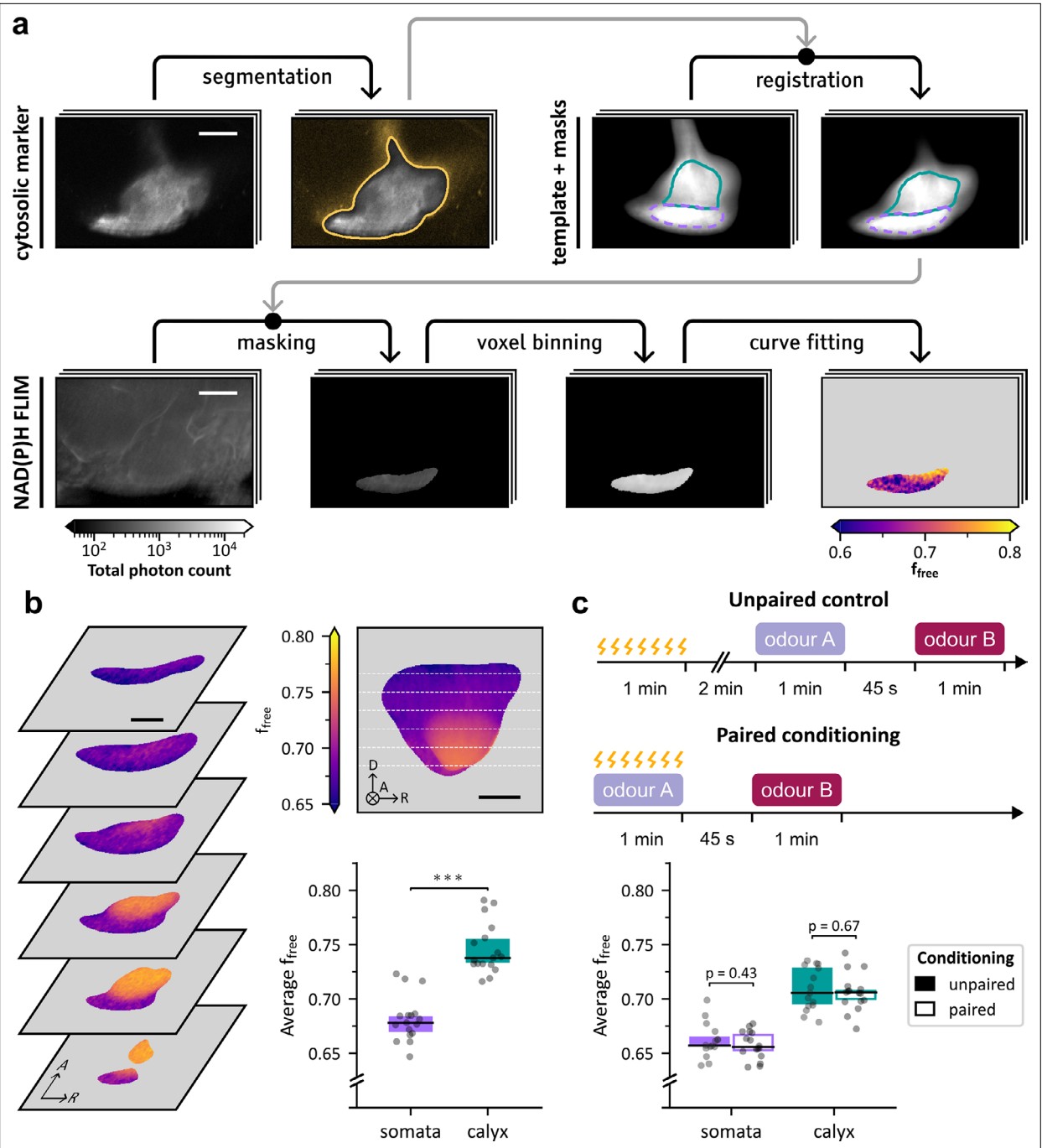

**Figure 3.** NAD(P)H states in the somata and calyx regions of the mushroom body (MB), for naive and conditioned flies. Scale bars: 25 μm. (**a**) Illustration of the main steps of the computation of the metabolic profile in the Kenyon cell (KC) somata region. The fluorescence intensity of the KC-specific cytosolic marker MB-DsRed was simultaneously recorded with the NAD(P)H fluorescence lifetime imaging (FLIM) image. The marker image was segmented using thresholding to isolate the MB. Next, we used a MB template with previously defined masks delineating the KC somata region (purple) and the calyx (turquoise). This template was registered to the segmented marker image. The registered masks could be used to isolate the target area (the somata region in this illustration) in the FLIM image. The masked FLIM image was spatially binned using a disk-shaped kernel (see Methods), and the resulting decays were fitted to obtain $f_{free}$ and $\tau_{bound}$ values in the target area. Each step is illustrated by a single horizontal section of the example 3D image. (**b**) Left: Horizontal sections of the average spatial distribution of $f_{free}$ in the somata area and calyx in naive flies. The average distribution was obtained by computing maps of $f_{free}$ in the posterior MB region, registering them to the MB template based on the corresponding MB-DsRed images and averaging the results. Axes indicate anterior (A), dorsal (D) and right (R) anatomical orientations. Top right: Average of $f_{free}$ across all transverse sections of the average spatial distribution, with white dashed lines indicating the depth of the horizontal sections. The circled cross indicates the direction pointing away from the viewer. Bottom right: Within-subject averages of $f_{free}$ in the somata and calyx regions. $f_{free}$ values in the somata

*Figure 3 continued on next page*

*Figure 3 continued*

region (n=17, μ=0.682±0.021) were significantly lower than in the calyx (n=17, μ=0.746±0.023; paired samples Student's *t*-test, $t_{16}$=16.3, $p$=2.1×10$^{-11}$). (**c**) Top: Schematic representation of the aversive conditioning protocol. Control flies underwent unpaired training, receiving electric shocks, followed by sequential exposure to odors A and B after a 2 min break. Conditioned flies underwent paired conditioning, receiving electric shocks while exposed to odor A, followed by exposure to odor B. Bottom: Within-subject averages of $f_{free}$ in the somata and calyx regions for conditioned flies. In the somata region, $f_{free}$ values for the flies subjected to unpaired stimuli (n=15, μ=0.662±0.016) were not significantly different than for the ones subjected to paired conditioning (n=16, μ=0.658±0.013; Student's *t*-test, $t_{29}$=0.8, $p$=0.43). In the calyx, $f_{free}$ values for the flies subjected to unpaired stimuli (n=15, μ=0.709±0.020) were not significantly different than for the ones subjected to paired conditioning (n=16, μ=0.706±0.017; Student's *t*-test, $t_{29}$=0.4, $p$=0.67).

The online version of this article includes the following figure supplement(s) for figure 3:

**Figure supplement 1.** Influence of memory conditioning on average $\tau_{bound}$ values in the somata and calyx region.

and γ subtypes (*Figure 4b*). Considering $\tau_{bound}$, no statistically significant difference was observed between the three subtypes (*Figure 4—figure supplement 1a*).

Contrarily to previously observed regional differences, KC subtypes differed on $f_{free}$ but not on $\tau_{bound}$. This suggests, in line with the established interpretation of $f_{free}$, that the basal metabolism of α/β neurons rely more heavily on glycolysis relative to OXPHOS than that of γ neurons. We conducted an initial investigation to determine whether the lower $f_{free}$ values observed in γ neurons could be attributed to a higher reliance on lactate-to-pyruvate conversion catalyzed by lactate dehydrogenase

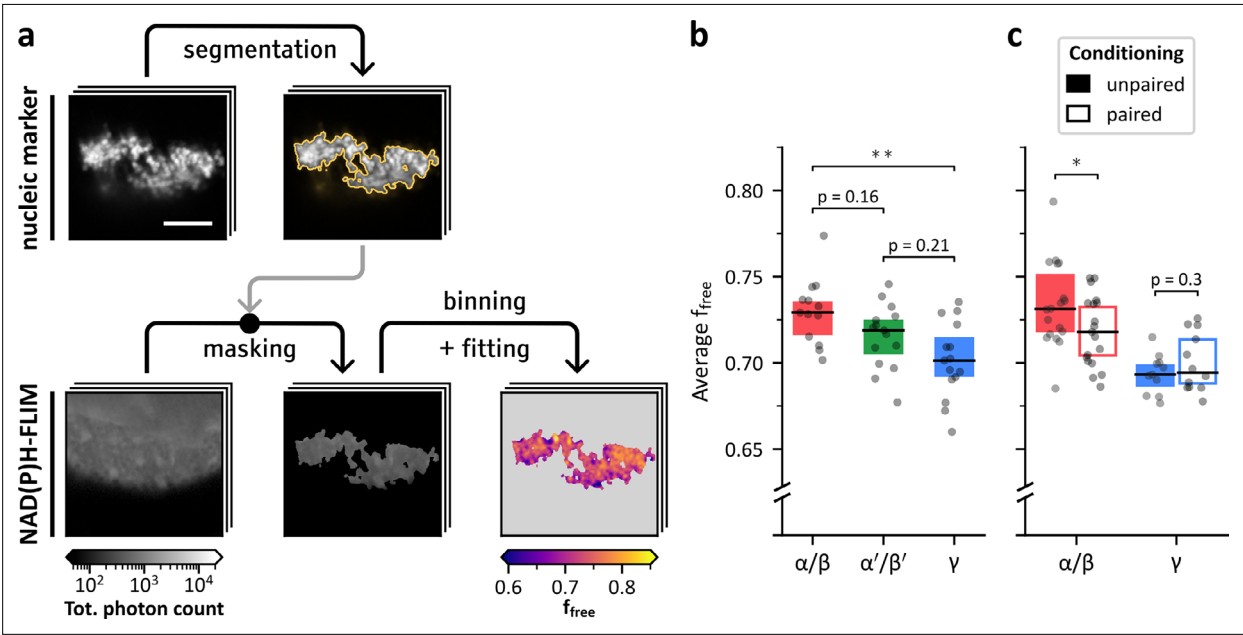

**Figure 4.** NAD(P)H states around somata of Kenyon cell (KC) subtypes in naive and conditioned flies. (**a**) Illustration of the main steps of the computation of the metabolic profile around the somata of a given KC subtype. Standard fluorescence intensity image of the subtype-specific nuclear marker was simultaneously recorded with the NAD(P)H fluorescence lifetime imaging (FLIM). The marker image was thresholded to remove background noise, and the resulting segmentation was used to mask the FLIM image. The masked FLIM image was spatially binned using a disk-shaped kernel (see Methods) and fitted to obtain $f_{free}$ values in the target area. Each step is illustrated by a single horizontal section of the example 3D image. Scale bar: 25 μm. (**b**) Within-subject averages of $f_{free}$ near the somata of α/β, α'/β', and γ KCs show significant variations (one-way ANOVA, $F_{2,40}$ = 7.1, $p$=2.3×10$^{-3}$). $f_{free}$ is significantly higher near α/β neurons (n=13, μ=0.730±0.019) than near γ neurons (n=15, μ=0.701±0.022; Student's *t*-test, $t_{26}$=3.6, Bonferroni-adjusted $P$=3.6×10$^{-3}$). The values of $f_{free}$ for α'/β' neurons (n=15, μ=0.715±0.018) were not significantly different from those near α/β (Student's *t*-test, $t_{26}$=2.0, Bonferroni-adjusted $p$=0.16) or γ neurons (Student's *t*-test, $t_{28}$=1.9, Bonferroni-adjusted $p$=0.21). (**c**) Within-subject averages of $f_{free}$ near the somata of α/β and γ KCs after odor conditioning. Near α/β somata, $f_{free}$ is significantly lower after paired conditioning (n=19, μ=0.718±0.020) compared to the control condition (n=18, μ=0.735±0.025; Student's *t*-test, $t_{35}$=2.0, $p$=4.99×10$^{-2}$). No statistically significant difference was observed near γ somata between conditioned (n=12, μ=0.700±0.017) and control flies (n=11, μ=0.693±0.011; Student's *t*-test, $t_{21}$=1.1, $p$=0.30).

The online version of this article includes the following figure supplement(s) for figure 4:

**Figure supplement 1.** Measures of $\tau_{bound}$ near the somata of Kenyon cell (KC) subtypes in naive and conditioned flies.

**Figure supplement 2.** Influence of Ldh knockdown on $f_{free}$ and $\tau_{bound}$ around the somata of γ neurons.

(Ldh) rather than pyruvate production by glycolysis, as Ldh was recently shown to be preferentially expressed in γ KCs as compared to other KC subtypes (*Raun et al., 2023*). However, we found no significant differences in f_{free} between Ldh-knockdown flies and controls (Dataset 7 in Methods; *Figure 4—figure supplement 2*).

This experiment provides initial insights into the metabolic heterogeneity of KC subtypes. Due to the limitations of the segmentation, it should be noted that our observations of the KC somata of a given subtype could partially include the surrounding glial processes.

## Memory formation induces a metabolic shift in α/β KCs

After establishing the basal metabolic profiles of KC subtypes, we sought to measure the influence of memory formation on specific subtypes. We imaged the MB cortical region in conditioned flies expressing subtype-specific nuclear markers (Dataset 8 and Dataset 9 in Methods). We focused on α/β and γ neurons, which have been reported to support the formation of MTM and STM, respectively, following a single training cycle. Our analysis showed that flies exposed to paired training exhibited a significant, albeit small-magnitude, decrease in f_{free} values around α/β somata compared with controls (*Figure 4c*). In contrast, the average f_{free} value in γ somata appeared unaffected by the type of conditioning. The effect of paired conditioning on $\tau_{bound}$ around α/β somata was not statistically significant (*Figure 4—figure supplement 1b*). We questioned whether summarizing changes in f_{free} values by a comparison of the within-subject averages was appropriate and if memory conditioning affected the shape of f_{free} distributions around α/β somata in another way than a shift of the mean, for instance, in the case of a shift concerning a neuronal subpopulation of α/β neurons. However, observation and

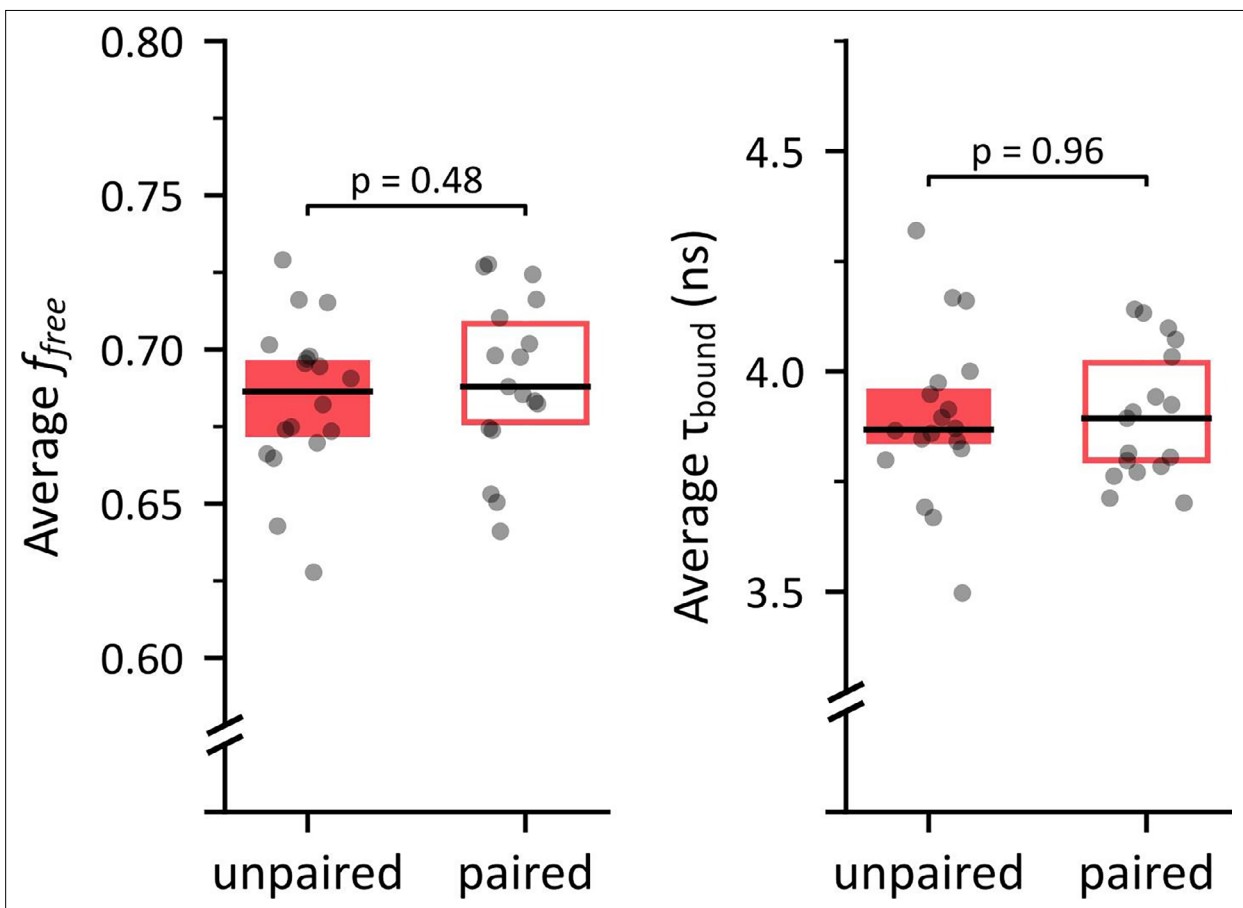

**Figure 5.** Influence of memory conditioning on average f_{free} and $\tau_{bound}$ values around α/β Kenyon cells (KCs) in ALAT-KD flies. Average decay parameter values near α/β somata in ALAT-KD mutant flies following unpaired versus paired memory conditioning. Left: Within-subject averages of f_{free}. No significant difference was observed between unpaired (n=18, μ=0.684±0.025) and paired conditions (n=17, μ=0.690±0.027; $t_{33}$=0.71, $p$=0.48). Right: Within-subject averages of $\tau_{bound}$. No significant difference was observed between unpaired (n=18, μ=3.90±0.19 ns) and paired conditions (n=17, μ=3.90±0.15 ns; $t_{33}$=0.05, $p$=0.96).

quantitative analysis of the $f_{free}$ distribution shapes confirmed that the metabolic shift in α/β somata significantly affects the mean value of $f_{free}$, the variance of free distribution was not affected, and its skewness did not show a significant difference either (Supplementary results in Appendix 1).

Recent findings show that MTM formation is coupled with increased mitochondrial activity in KCs relying on the conversion to pyruvate of alanine supplied by glia (*Rabah et al., 2023*). This conversion being catalyzed by alanine aminotransferase (ALAT), measured the influence of conditioning in mutant flies expressing ALAT RNAi in α/β neurons (Dataset 10 in Methods). We observed no learning-induced effect on $f_{free}$ in this data (*Figure 5* and *Appendix 1—figure 11*), which is coherent with the possibility that the previously observed decrease of $f_{free}$ in flies is due to increased mitochondrial activity supported by alanine. It should, however, be noted that the average $f_{free}$ values obtained for ALAT-KD flies in the unpaired and paired conditions (μ=0.684±0.025 and μ=0.690±0.027, respectively) were noticeably lower than those previously obtained in flies with undisturbed ALAT expression (Dataset 9; 0.718±0.020 and μ=0.735±0.025, respectively). This difference suggests that ALAT knockdown in α/β neurons may alter their basal metabolic profile.

These experiments demonstrated the capability of NAD(P)H FLIM to detect subtle physiological alterations of cellular metabolism induced by memory formation. Moreover, the results indicate that distinct metabolic dynamics unfold in α/β and γ neurons after aversive conditioning. More specifically, the decrease of $f_{free}$ measured for α/β neurons suggests that, during memory formation, ATP is increasingly produced through OXPHOS relative to glycolysis.

## Discussion

In this study, we sought to demonstrate the potential of NAD(P)H FLIM for detecting physiologically relevant spatiotemporal dynamics in brain energy metabolism. By applying this technique to the central brain of *Drosophila melanogaster*, we uncovered new insights into cellular metabolic heterogeneity. To establish this novel application, we employed bi-exponential fitting to extract two key parameters: the free fraction of NAD(P)H ($f_{free}$) and the lifetime of enzyme-bound NAD(P)H ($\tau_{bound}$). Using genetically encoded anatomical markers combined with registration tools, these parameters were measured across different brain regions.

On a broad scale, our findings revealed the distinct metabolic profiles of ensheathing glial layers, cortical areas, and neuropil regions, with $\tau_{bound}$ being particularly discriminative of the neuropil and cortex. At the MB level, differences between KC somata and calyx regions were observed in both $\tau_{bound}$ and $f_{free}$. At a finer scale, significant basal and memory-induced differences were detected in $f_{free}$ among KC subtypes, whereas changes in $\tau_{bound}$ remained limited. These patterns suggest that tissue-level differences are primarily reflected by $\tau_{bound}$, whereas subtype-specific variations among KCs are predominantly captured by $f_{free}$.

As FLIM-based metabolic measurements are influenced by numerous factors (*Schaefer et al., 2019*), the interpretation of these parameters in this novel context remains multifaceted. While $f_{free}$ is often considered a reliable indicator of the glycolysis-to-OXPHOS balance, $\tau_{bound}$ is thought to reflect a broader range of factors, including the composition of proteins binding to NAD(P)H and the NADPH/NADH ratio. The tissue-level differences dominated by $\tau_{bound}$, such as those observed between the posterior MB and other regions (*Figure 1e*), likely reflect broad metabolic distinctions that extend beyond energy-related processes. In contrast, the differences in $f_{free}$ between KC subtypes may more reliably indicate distinct reliance on glycolysis versus OXPHOS. Instances where both $f_{free}$ and $\tau_{bound}$ differ strongly – such as between the KC somata and calyx (*Figure 3b*, *Figure 3—figure supplement 1a and b*) – might result from a combination of these factors. Measures of $f_{free}$ and $\tau_{bound}$ in the posterior part of the MB and within KC subtypes were notably anticorrelated (*Appendix 1—figure 3*). We found, via simulation, that this anticorrelation was not an artifact of the fitting process, supporting the existence of a biological dependence between both parameters in our data (Supplementary results in Appendix 1).

At the level of KC subtypes, we found that α/β neurons exhibited higher $f_{free}$ values than γ neurons, suggesting a higher glycolysis-to-OXPHOS ratio in α/β neurons. Ldh has been reported to be more expressed in γ neurons compared to the other KC subtypes (*Raun et al., 2023*), leading us to hypothesize that lactate-to-pyruvate conversion by Ldh reduces the glycolytic demand in this subtype. A knockdown of Ldh in γ neurons was conducted as a preliminary investigation, but no significant impact on $f_{free}$ was observed. Disentangling the contributions of KC somata and surrounding cortex glia to

the FLIM signal, potentially through co-labeling approaches, could deepen our understanding of the basal metabolic heterogeneity among KC subtypes. However, current technical limitations make such experiments challenging. Cortex glia processes are very thin and closely surround neuronal somata, making them difficult to resolve separately. In addition, residual preparation movement and the extensive number of excitations required to collect sufficient photons in the FLIM channel hinder reliable separation of glial and neuronal signals.

Notably, we observed a decrease in $f_{free}$ in α/β somata – and surrounding glial processes – following a single aversive conditioning session, with no corresponding change in γ neurons. This reduction may reflect a transition towards OXPHOS in α/β somata during memory formation or maintenance. Although single-session memory training primarily induces the encoding of labile STM in γ neurons, parallel processes in α/β neurons lead to the formation of longer-persisting MTM (*Blum et al., 2009*; *Séjourné et al., 2011*; *Bouzaiane et al., 2015*). Our observation of a metabolic shift specific to α/β neurons aligns with recent evidence showing increased mitochondrial activity in KCs during MTM formation, supported by alanine transfer from glia (*Rabah et al., 2023*). We found that this shift was not detected in ALAT-KD flies, a first step towards the validation that the observed shift relies on alanine consumption.

The magnitude of the memory-induced metabolic shift observed in α/β neurons appears modest relative to the variability of $f_{free}$ measurements. This underscores the need for further investigations to confirm the existence of this shift, its amplitude, and to clarify the underlying mechanisms. Our pioneer investigation thus provides a lower boundary for the sample size that should be envisaged when using NAD(P)H FLIM in vivo to detect experience-dependent metabolic variations. Several factors may potentially contribute to an underestimation of the true magnitude of the shift. First, the inherently probabilistic nature of memory formation in *Drosophila* could dilute the average effect, as aversive olfactory conditioning may induce changes of variable magnitude across individuals (*Quinn et al., 1974*; *Tully and Quinn, 1985*). As we could not technically evaluate the memory performance of flies during or after metabolic imaging, we unfortunately could not assess the correlation, at the individual level, between $f_{free}$ values and the strength of memory at the behavioural level. Second, broader fluctuations in neuronal metabolic profiles – such as those driven by circadian rhythms (*Xu et al., 2008*) – are largely corrected for through the use of control groups, but may still contribute to increased variability in the measured indicators (*Stringari et al., 2015*). Characterization of these fluctuations could provide a reference to evaluate the magnitude of conditioning-induced shifts and guide experimental design to improve the sensitivity of future measurements. Finally, the presence of surrounding cortex glia, with their own metabolic dynamics, may contribute to the signal and thereby obscure shifts occurring specifically within neurons.

While interpretations of the observed NAD(P)H state variations in this new context remain open-ended, our findings lay a solid foundation for developing hypotheses about energy regulation in the MBs. This novel application of NAD(P)H FLIM can be extended in numerous ways to probe the metabolic dynamics of memory formation across different conditions, resolutions, and timescales. This approach also holds promise for investigating the role of metabolic dysregulation in age-related cognitive decline and neurodegenerative diseases (*Saitoe et al., 2005*; *Camandola and Mattson, 2017*). More broadly, it offers a powerful tool to advance fundamental neuroenergetic research, deepening our understanding of the interplay between energy metabolism and brain function.

# Materials and methods

**Key resources table**

| Reagent type (species) or resource | Designation | Source or reference | Identifiers | Additional information |
|---|---|---|---|---|
| Genetic reagent (*D. melanogaster*) | MB010B | BDSC | RRID:BDSC_68293 | |
| Genetic reagent (*D. melanogaster*) | MB008B | BDSC | RRID:BDSC_68291 | |
| Genetic reagent (*D. melanogaster*) | MB005B | BDSC | RRID:BDSC_68306 | |

*Continued on next page*

*Continued*

| Reagent type (species) or resource | Designation | Source or reference | Identifiers | Additional information |
|---|---|---|---|---|
| Genetic reagent (*D. melanogaster*) | MB009B | BDSC | RRID:BDSC_68292 | |
| Genetic reagent (*D. melanogaster*) | UAS-Stinger | BDSC | RRID:BDSC_84277 | |
| Genetic reagent (*D. melanogaster*) | UAS-mCherry-NLS | BDSC | RRID:BDSC_38424 | |
| Genetic reagent (*D. melanogaster*) | UAS-Ldh$^{RNAi}$ (HMS00039) | BDSC | RRID:BDSC_33640 | |
| Genetic reagent (*D. melanogaster*) | UAS-ALAT$^{RNAi}$ (GD9174) | VDRC | RRID:Flybase_FBst0459715; VDRC Id 32681 | |
| Genetic reagent (*D. melanogaster*) | MB-DsRed | *Riemensperger et al., 2005* | | |
| Chemical compound, drug | 3-octanol (99%) | Sigma-Aldrich | Cat. #153095 | |
| Chemical compound, drug | 4-methylcyclohexanol (98%) | Sigma-Aldrich | Cat. #218405 | |
| Chemical compound, drug | Paraffine GPR Rectapur | VWR | Cat. #24679.360 | |
| Chemical compound, drug | NaCl | Sigma-Aldrich | Cat. #S9625 | |
| Chemical compound, drug | KCl | Sigma-Aldrich | Cat. #P3911 | |
| Chemical compound, drug | MgCl$_2$ | Sigma-Aldrich | Cat. #M9272 | |
| Chemical compound, drug | CaCl$_2$ | Sigma-Aldrich | Cat. #C3881 | |
| Chemical compound, drug | D-trehalose | Sigma-Aldrich | Cat. #9531 | |
| Chemical compound, drug | Sucrose | Sigma-Aldrich | Cat. #S9378 | |
| Chemical compound, drug | HEPES-NaOH | Sigma-Aldrich | Cat. #H7637 | |
| Software, algorithm | Leica Application Suite X (v3.5.7) | Leica Microsystems | RRID:SCR_013673 | |
| Software, algorithm | ANTs | http://www.picsl.upenn.edu/ANTS/ | RRID:SCR_004757 | |

## Fly strains

Flies (*Drosophila melanogaster*) were raised on standard fly food (inactivated yeast 6% w/v; corn flour 6.66 % w/v; agar 0.9% w/v; methyl 4-hydroxybenzoate 22 mM) on a 12 hr light/dark cycle at 18 °C with 60% humidity (unless mentioned otherwise). All experiments in this study were performed on 1–4 day-old adult female flies. Female flies were preferred for imaging experiments because of their larger size. The following split-GAL4 driver lines (*Aso et al., 2014*) were used to target specific populations of KCs: MB010B (all KCs; RRID:BDSC_68293), MB008B (α/β KCs; RRID:BDSC_68291), MB005B (α'/β' KCs; RRID:BDSC_68306), MB009B (γ KCs; RRID:BDSC_68292). Reporter lines were UAS-Stinger (RRID:BDSC_84277), UAS-mCherry-NLS (RRID:BDSC_38424; *Delestro et al., 2020*), UAS-Ldh$^{RNAi}$ (HMS00039; RRID:BDSC_33640), and UAS-ALAT$^{RNAi}$ (GD9174, third chromosome insertion; RRID:Flybase_FBst0459715), as well as MB-DsRed (*Riemensperger et al., 2005*).

## Olfactory conditioning

Olfactory conditioning was conducted similarly to other studies from the same research group (*Pascual and Préat, 2001*; *Isabel et al., 2004*; *Plaçais et al., 2012*; *Rabah et al., 2023*). Experimental flies were transferred to fresh bottles containing standard medium on the day before conditioning. Groups of ~30 flies were subjected to a single cycle olfactory conditioning protocol, using either paired or unpaired (control) stimuli in custom-made barrel-type machines (*Pascual and Préat, 2001*). Throughout the conditioning protocol, each barrel was attached to a constant air flow at 2 l/min. For a single cycle of associative training, flies were first exposed to an odorant A for 1 min while twelve 60-Volt 1.25 s square-wave pulses were delivered at 5 s intervals; flies were then exposed 45 s

later to a second odorant B without shocks for 1 min (*Figure 3c*, top). The odorants 3-octanol and 4-methylcyclohexanol, diluted in paraffin oil at a final concentration of $2.79 \cdot 10^{-1}$ g/l, were used as odorants A and B, respectively. During unpaired conditionings, the shock and odor stimuli were delivered sequentially, with the first odor being released 2 min after the end of the shocks.

## Image acquisition

Data were collected indiscriminately from 30 min to 1.5 hr after memory training. The different experimental groups constituting each dataset were imaged during the same experimental sessions. A single fly was picked and prepared for imaging as previously described (*Plaçais et al., 2017*). Briefly, the fly was glued on a plastic slide using a biocompatible dental glue (3 M ESPE Protemp) without any prior anesthesia. An alignment wire was used to maintain the *Drosophila* head in a correct position. The head capsule was opened and the brain was exposed by gently removing the superior tracheae. The head capsule was bathed in artificial hemolymph solution for the duration of the preparation. The composition of this solution was NaCl 130 mM (Sigma cat. no. S9625), KCl 5 mM (Sigma cat. no. P3911), MgCl2 2 mM (Sigma cat. no. M9272), CaCl2 2 mM (Sigma cat. no. C3881), D-trehalose 5 mM (Sigma T cat. no. 9531), sucrose 30 mM (Sigma cat. no. S9378), and HEPES hemisodium salt 5 mM (Sigma cat. no. H7637). At the end of surgery, any remaining solution was absorbed and a fresh 90 µl droplet of this solution was applied on top of the brain.

Two-photon fluorescence lifetime measurements were performed using a Leica Microsystems SP8 DIVE-FALCON microscope equipped with a HC-IRAPO 25 x, 1.0 NA water-immersion objective coupled to a dual output InSight infrared excitation laser (Spectra Physics). Two-photon excitation of NAD(P)H autofluorescence was achieved using the tunable laser beam at 740 nm. Fluorescence was measured by time-correlated single-photon counting in the spectral range 420–500 nm. Fluorescent anatomical markers were imaged simultaneously. Ds-Red and mCherry were excited at 740 nm and their emission intensity was recorded in the spectral range 600–640 nm. GFP was excited at 910 nm and its emission intensity was recorded in the spectral range 500–550 nm. All data acquisition and analysis was performed with the Leica LAS X (v3.5.7) software. 3D z-stacks were acquired either on the whole brain or on the cortex region encompassing all Kenyon cells. The number of planes in the z-dimension was adjusted to include all visible KCs somata. The excitation laser intensity was adjusted with LAS X controls to 14% corresponding to ~9 mW at the objective output. On each x-y plane, 30 sequential scans were averaged, yielding on average 150–300 photons per pixel.

## Datasets

This section describes the different image datasets used throughout the present study.

### Dataset 1

Dataset 1 consists of five images from five flies expressing a KC-specific cytosolic marker (MB-DsRed). This marker was imaged simultaneously with NAD(P)H FLIM. Each 3D image captured the entire central brain, with an average resolution of 0.74 µm for the X-Y axes and 2.00 µm for the Z axis.

### Dataset 2

Dataset 2 consists of 31 images from 17 flies expressing a KC-specific cytosolic marker (MB-DsRed). This marker was imaged simultaneously with NAD(P)H FLIM. Each 3D image captured the somata and calyx region of a single hemisphere, with an average resolution of 0.40 µm for the X-Y axes and 2.00 µm for the Z axis. For additional details, see *Appendix 2—table 1*.

### Dataset 3

Dataset 3 consists of 165 images from 102 flies expressing a KC-specific cytosolic marker (MB-DsRed) and a subtype-specific nuclear marker (MB008B>mCherry::NLS; MB009B>mCherry::NLS; MB005B>mCherry::NLS). Both markers were simultaneously imaged in separate channels. Each 3D image captured the somata and calyx region of a single hemisphere, with an average resolution of 0.35 µm for the X-Y axes and 0.60 µm for the Z axis. For additional details, see *Appendix 2—table 2*.

### Dataset 4

Dataset 4 consists of 32 images from 19 flies expressing a KC-specific nuclear (MB010B>Stinger) and cytosolic (MB-DsRed) markers. Both markers were simultaneously imaged in separate channels. Each 3D image captured the somata and calyx region of a single hemisphere, with an average resolution of 0.38 μm for the X-Y axes and 0.60 μm for the Z axis. For additional details, see *Appendix 2—table 3*.

### Dataset 5

Dataset 5 consists of 31 images from 31 conditioned flies expressing a KC-specific cytosolic marker (MB-DsRed). This marker was imaged simultaneously with NAD(P)H FLIM. Each 3D image captured the somata and calyx region of a single hemisphere, with an average resolution of 0.39 μm for the X-Y axes and 2.00 μm for the Z axis. For additional details, see *Appendix 2—table 4*.

### Dataset 6

Dataset 6 consists of 79 images from 43 flies expressing a subtype-specific nuclear marker (MB008B>mCherry::NLS; MB009B>mCherry::NLS; MB005B>mCherry::NLS). This marker was imaged simultaneously with NAD(P)H FLIM. Each 3D image captured the somata and calyx region of a single hemisphere, with an average resolution of 0.38 μm for the X-Y axes and 4.00 μm for the Z axis. For additional details, see *Appendix 2—table 5*.

### Dataset 7

Dataset 7 consists of 24 images from 24 flies, with and without Ldh-knockdown mutations (MB009B>Ldh RNAi) in γ neurons, expressing a γ-specific nuclear marker (MB009B>mCherry::NLS). This marker was imaged simultaneously with NAD(P)H FLIM. Each 3D image captured the somata and calyx region of a single hemisphere, with an average resolution of 0.37 μm for the X-Y axes and 2.00 μm for the Z axis. For additional details, see *Appendix 2—table 6*.

### Dataset 8

Dataset 8 consists of 40 images from 38 conditioned flies expressing an α/β-specific nuclear marker (MB008B>mCherry::NLS). This marker was imaged simultaneously with NAD(P)H FLIM. Each 3D image captured the somata and calyx region of a single hemisphere, with an average resolution of 0.38 μm for the X-Y axes and 2.00 μm for the Z axis. For additional details, see *Appendix 2—table 7*.

### Dataset 9

Dataset 9 consists of 23 images from 23 conditioned flies expressing a γ-specific nuclear marker (MB009B>mCherry::NLS). This marker was imaged simultaneously with NAD(P)H FLIM. Each 3D image captured the somata and calyx region of a single hemisphere, with an average resolution of 0.39 μm for the X-Y axes and 2.00 μm for the Z axis. For additional details, see *Appendix 2—table 8*.

### Dataset 10

Dataset 10 consists of 23 images from 23 conditioned flies expressing an α/β-specific nuclear marker (MB008B>mCherry::NLS). This marker was imaged simultaneously with NAD(P)H FLIM. Each 3D image captured the somata and calyx region of a single hemisphere, with an average resolution of 0.39 μm for the X-Y axes and 2.00 μm for the Z axis. For additional details, see *Appendix 2—table 9*.

## Segmentation of fluorescence intensity images for FLIM masking

For Dataset 1, each image was initially divided into two halves along a manually defined median plane, enabling separate processing of both hemispheres. The somata and calyx region was roughly isolated from each image using manually defined planes orthogonal to the Y axis. A binary mask of this region was then obtained through Li thresholding (*Li and Tam, 1998*) after Gaussian smoothing (with a standard deviation σ=4 μm). In each single-hemisphere image, the entire MB was delineated by applying Li thresholding after Gaussian smoothing (σ=2 μm). The peduncle and lobes were first roughly isolated using manually defined planes, orthogonal to the main axes. Binary masks for these regions were then created using smoothing (σ=2 μm) and Otsu thresholding (*Otsu, 1979*). Finally, all single-hemisphere masks were combined to reconstruct complete brain masks.

Images of Dataset 2 and Dataset 5 were segmented by applying Li thresholding after Gaussian smoothing (σ=2 μm) and keeping only the largest contiguous volume.

Images of Dataset 6, Dataset 7, Dataset 8, and Dataset 9 were segmented in two steps. First, Li thresholding was applied after Gaussian smoothing (σ=0.5 μm) and only the volumes superior to 1 μm³ were kept. Then, Li thresholding was applied again within the resulting non-zero voxels.

## FLIM image processing

We observed that voxels with exceptionally high photon counts were predominantly associated with tracheas or with objects outside the region of interest due to the limitations of the segmentations. To reduce the influence of these artifacts while preserving the neuronal signal, we excluded voxels with photon counts exceeding $Q_3+3\times IQR$, where $Q_3$ represents the upper quartile of the count distribution and $IQR$ denotes the interquartile range.

To ensure that the total photon counts of the decay histograms were sufficient for reliable curve fitting, the signals of neighboring voxels were combined. This process, known as spatial binning, enhances the signal-to-noise ratio of the decay signal at the cost of reducing spatial resolution. Decay histograms were summed within a disk centered on each voxel. This disk was defined as a 5×5 ×1 voxel volume, with the four corner voxels excluded. To avoid unreliable modeling of the decays, voxels for which the total photon count did not exceed 500 after binning were excluded from further analyses.

Theoretical fluorescence decays were modeled using the following bi-exponential function $D$:

$$d(t) = f_{free} \, exp\left(-\frac{t}{\tau_{free}}\right) + \left(1 - f_{free}\right) exp\left(-\frac{t}{\tau_{bound}}\right)$$

where $f_{free}$ is the free NAD(P)H fraction, $\tau_{free}$ is the free NAD(P)H lifetime and $\tau_{bound}$ is the protein-bound NAD(P)H lifetime. The measured decays were fitted with the following function $f$, which incorporates the theoretical decay and the effects of the measurement process:

$$f(t) = \sum_{p=0}^{1} \left[ d(t) * IRF\left(t - \Delta t + iT\right)\right]$$

where $IRF$ is the microscope impulse response function, $\Delta t$ is the temporal shift between the IRF and the measured signal and $T$ is the laser period. The modeled signal at time $t$ takes into account the fluorescence caused by the last laser pulse (term for $p=0$) as well as the previous one (term for $p=1$), as proposed by *Klemm et al., 2015*.

In practice, $f$ was evaluated for $N_t$ = 129 time steps $t_i$ corresponding to the time bins of the FLIM data and normalized:

$$f_i = \frac{f\left(t_i\right)}{\sum_{j=1}^{Nt} f\left(t_j\right)}$$

The final simulated decay sequence $s$ was defined as:

$$s_i = \frac{f_i + b}{\sum_{j=1}^{N_t} \left(f_j + b\right)}$$

where $b$ is a constant background noise.

The IRF was obtained from each image using the FALCON module of the Leica LAS X software managing FLIM acquisitions (v3.5.7). $T$ was set to 12.5 ns, corresponding to the 80 MHz repetition rate of the microscope laser. Due to its limited physiological interpretability and its narrow reported range of variation (*Guo et al., 2008*), $\tau_{free}$ was fixed at 0.4 ns, in line with values reported in the literature for aqueous solutions (*Scott et al., 1970*; *Visser and Hoek, 1981*), cell cultures (*Wu et al., 2006*; *Rück et al., 2014*; *Sharick et al., 2018*), tissue slices (*Vishwasrao et al., 2005*), in vivo experiments (*Yaseen et al., 2013*), as well as in our own findings (Supplementary results in Appendix 1).

Parameters $f_{free}$, $\tau_{bound}$, $\Delta t$, and $b$ were estimated via maximum-likelihood estimation (MLE), using the 2*I** quality parameter (*Maus et al., 2001*):

$$2I^* = 2 \sum_{i=1}^{N_t} m_i ln \left( \frac{m_i}{Cs_i} \right)$$

where $m_i$ represents the measured photon count in bin i and C represents the total photon count of the measured decay (i.e. the sum of $m_i$ values). MLE was preferred over the least squares method for its higher reported accuracy (*Maus et al., 2001*; *Laurence and Chromy, 2010*; *Chen et al., 2022*).

This fitting process was implemented using the Nelder-Mead algorithm (*Gao and Han, 2012*). Each curve fitting was repeated 5 times, using initialization parameters uniformly sampled from empirically defined ranges. The parameter estimation having returned the lowest 2I* value was selected. Voxels with final 2I* values exceeding the empirically defined threshold of 200 were systematically excluded from further analysis (*Figure 1—figure supplement 1*).

To accelerate the estimation for all voxels of an image, a two-step process was carried out. An initial fitting was computed on a downsampled version of the image, which had been decimated by a factor of 3 in the X and Y directions. The time shift $\Delta t$ was then fixed to the median of the obtained values, enabling the application of a streamlined three-parameter model to the complete image.

In the end, we considered that the metabolic profile extracted from a given decay curve is constituted of the couple ($f_{free}$, $\tau_{bound}$). The mean lifetime was optionally computed as the sum of the free and bound NAD(P)H lifetimes weighted by their fractional contributions:

$$\tau_{mean} = f_{free}\tau_{free} + \left( 1 - f_{free} \right) \tau_{bound}$$

Analysis of the spatial distributions revealed that voxels with $f_{free}$ values below 0.2 consistently fell outside the regions of interest as a result of imperfect segmentation. Consequently, these voxels were systematically excluded from further analysis.

## Building a template image of the somata and calyx region

A template image of the somata region of the MB was generated from the 165 KC-specific cytosolic marker channel of the images (Dataset 3). The preprocessing steps for these images are outlined in *Figure 2a*. Images were resampled to an isotropic voxel size of 0.5 μm in all directions. To eliminate background noise and extraneous bright objects, we applied Li thresholding to a smoothed version of each image, retaining only the brightest contiguous object. The images were cropped and padded to make their size uniform and to center them based on their center of mass. The average intensity histogram of the dataset was computed by averaging the intensity histograms of all images. The intensity of all images were normalized by adjusting their intensity histogram to match the dataset's average histogram (*Nyúl et al., 2000*).

The template was subsequently constructed from the preprocessed images using the symmetric group-wise normalization algorithm (*Avants et al., 2010*). The registration method used in the procedure was defined as a sequence of rigid, affine, and diffeomorphic transformations applied to the 3D volumetric images. Symmetric normalization (*Avants et al., 2008*) was used as the diffeomorphic transformation, with empirically defined parameters of 0.2 for the gradient step, 3 for the update field smoothing, and 1 for the displacement field smoothing. The method initializes the template as the mean image and then iteratively updates it to minimize the average displacement required to register the dataset to the template (*Figure 2b*). The template image associated with the lowest average displacement after eight iterations was selected. The template was finally refined by applying thresholding and centering. The final template image is shown in *Figure 2cd*.

## Building masks for the somata and calyx region

A mask isolating the somata region was derived from the images of Dataset 4, obtained by simultaneously capturing KC-specific nuclear and cytosolic markers on two separate channels. This set of images was registered to the previously established MB template (*Figure 2—figure supplement 1*) using the cytosolic marker channel. The nuclear marker channels were warped using the obtained transformations and then averaged. The somata region mask was obtained by binarizing the result. The binarization used the highest threshold obtained through the application of the 3-class multi-Otsu algorithm (*Liao et al., 2001*) after Gaussian smoothing (σ=3 μm).

A mask isolating the calyx region was derived by binarizing the MB template image. The binarization used the highest threshold obtained through the application of the 3-class multi-Otsu algorithm

after Gaussian smoothing (σ=5 μm). The final calyx mask was refined by excluding the voxels included in the somata mask.

## Mapping $f_{free}$ in the somata and calyx region

Single-hemisphere maps of $f_{free}$ were obtained from Dataset 2. Right-side images were mirrored to allow their superposition with left hemisphere images. The MB template was registered to each image. This transformation was applied to the chosen mask (somata or calyx) using nearest-neighbor interpolation. The warped mask was applied to select the region of interest in the FLIM image. Spatial binning and curve fitting was applied to the masked FLIM image, resulting in maps of the decay parameters over the region of interest.

To map decay parameters in the template space, the KC-specific marker channel (MB-DsRed) of the image was registered to the template. This deformation was applied to the maps of decay parameters using nearest-neighbor interpolation. The average $f_{free}$ map presented in *Figure 3b* was obtained by averaging all registered parameters maps.

## Statistics

All indicated replicates correspond to biological replicates. For all boxplots, whiskers show the minimum and maximum values, the box shows the first and third quartiles, and the horizontal line displays the median. All tested distributions are described by their sample size (n=sample size) and their mean and standard deviation (μ=men ± standard deviation). All Student's *t*-tests are two-tailed independent samples *t*-tests, unless stated otherwise. In the case of multiple comparisons, displayed and reported *p*-values are the Bonferroni-adjusted *p*-values, as specified in the corresponding captions.

## Code availability

The code used to process marker and FLIM images is available as a GitHub repository https://github.com/biocompibens/MB-FLIM, copy archived at *Philémon and biocompibens, 2025*.

## Acknowledgements

This research has received support from the Region Ile de France through the DIM Elicit program, from the Institut Convergences QLife (ANR-17-QLIFE), from the LabEx MemoLife (ANR-10-LABX-54 MEMOLIFE and ANR-10-IDEX-0001-02 PSL* Université Paris), from the European Research Council (ERC-AdG-741550), from the Human Frontier Science Program (RGY0078/2017 ChroMet), and from the Agence Nationale de la Recherche (ANR) under contracts ANR-10-INBS-04 France BioImaging and ANR-11-EQPX-0029 Morphoscope2.

## Additional information

### Funding

| Funder | Grant reference number | Author |
| --- | --- | --- |
| Agence Nationale de la Recherche | ANR-17-QLIFE | Philémon Roussel |
| Agence Nationale de la Recherche | ANR-10-LABX-54 MEMOLIFE | Philémon Roussel |
| Agence Nationale de la Recherche | ANR-10-IDEX-0001-02 PSL* | Auguste Genovesio |
| Agence Nationale de la Recherche | ANR-10-INBS-04 | Chiara Stringari |
| Agence Nationale de la Recherche | ANR-11-EQPX-0029 | Chiara Stringari |
| European Research Council | ERC-AdG-741550 | Thomas Preat |

| Funder | Grant reference number | Author |
|---|---|---|
| Human Frontier Science Program | RGY0078/2017 ChroMet | Chiara Stringari |
| Région Ile-de-France | DIM ELICIT | Pierre-Yves Plaçais Auguste Genovesio |

The funders had no role in study design, data collection and interpretation, or the decision to submit the work for publication.

## Author contributions
Philémon Roussel, Conceptualization, Data curation, Software, Formal analysis, Investigation, Visualization, Methodology, Writing – original draft, Writing – review and editing; Mingyi Zhou, Investigation; Chiara Stringari, Thomas Preat, Conceptualization, Writing – review and editing; Pierre-Yves Plaçais, Conceptualization, Data curation, Supervision, Funding acquisition, Investigation, Methodology, Writing – original draft, Project administration, Writing – review and editing; Auguste Genovesio, Conceptualization, Formal analysis, Supervision, Funding acquisition, Investigation, Methodology, Writing – original draft, Project administration, Writing – review and editing

## Author ORCIDs
Philémon Roussel ⬤ https://orcid.org/0009-0009-6377-2167
Mingyi Zhou ⬤ https://orcid.org/0000-0002-4544-1489
Chiara Stringari ⬤ https://orcid.org/0000-0002-0550-7463
Pierre-Yves Plaçais ⬤ https://orcid.org/0000-0001-8426-4465
Auguste Genovesio ⬤ https://orcid.org/0000-0003-1877-5595

Reviewer #1 (Public review): https://doi.org/10.7554/eLife.106040.3.sa1
Reviewer #2 (Public review): https://doi.org/10.7554/eLife.106040.3.sa2
Reviewer #3 (Public review): https://doi.org/10.7554/eLife.106040.3.sa3
Author response https://doi.org/10.7554/eLife.106040.3.sa4

# Additional files

## Supplementary files
MDAR checklist

## Data availability
Data, including FLIM, marker imaging and IRFs, is available in the BioStudies database under accession number S-BIAD1528.

The following dataset was generated:

| Author(s) | Year | Dataset title | Dataset URL | Database and Identifier |
|---|---|---|---|---|
| Roussel P, Zhou M, Stringari C, Preat T, Plaçais PY, Genovesio A | 2025 | In vivo autofluorescence lifetime imaging of the Drosophila brain captures metabolic shifts associated with memory formation | https://www.ebi.ac.uk/biostudies/bioimages/studies/S-BIAD1528 | BioStudies, S-BIAD1528 |

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

# Appendix 1

## Supplementary results

### The template image minimizes displacement upon registration

We confirmed that registering the images to the template image resulted in less displacement compared to registration to either the average image or to randomly selected images (*Appendix 1—figure 1*). This finding supports the use of the template image as a reference space to minimize deformation during registration.

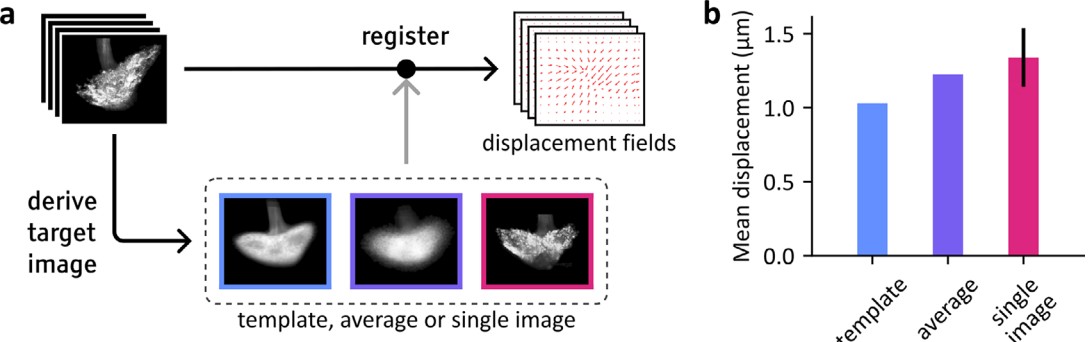

**Appendix 1—figure 1.** Validation of the mushroom body (MB) template. (**a**) Schematic illustrating the process of evaluation of the amount of displacement required to match MB images to different target images, namely the template, the raw average, or a randomly selected image. (**b**) Mean displacements resulting from the registration of all images of the dataset to different target images. Registration to a single image was performed 5 times on different randomly chosen images.

### Minimal differences in hemisphere-specific MB shapes

We compared templates generated from left, right, or both hemispheres images of the somata region of the MBs (*Appendix 1—figure 2a*; Supplementary methods). Differences between left- and right-specific template images were consistently higher than differences obtained using mixed hemispheres (*Appendix 1—figure 2b*). As the subset size increased, this effect became stronger. For subsets of 30 images, the average differences were 11.7% and 14.4% for inter-hemisphere and mixed-hemisphere comparisons, respectively. This result suggests the existence of a slight difference in shape between left and right MBs. We concluded upon visual inspection that this putative difference was too subtle to justify the use of separate templates (*Appendix 1—figure 2c and d*).

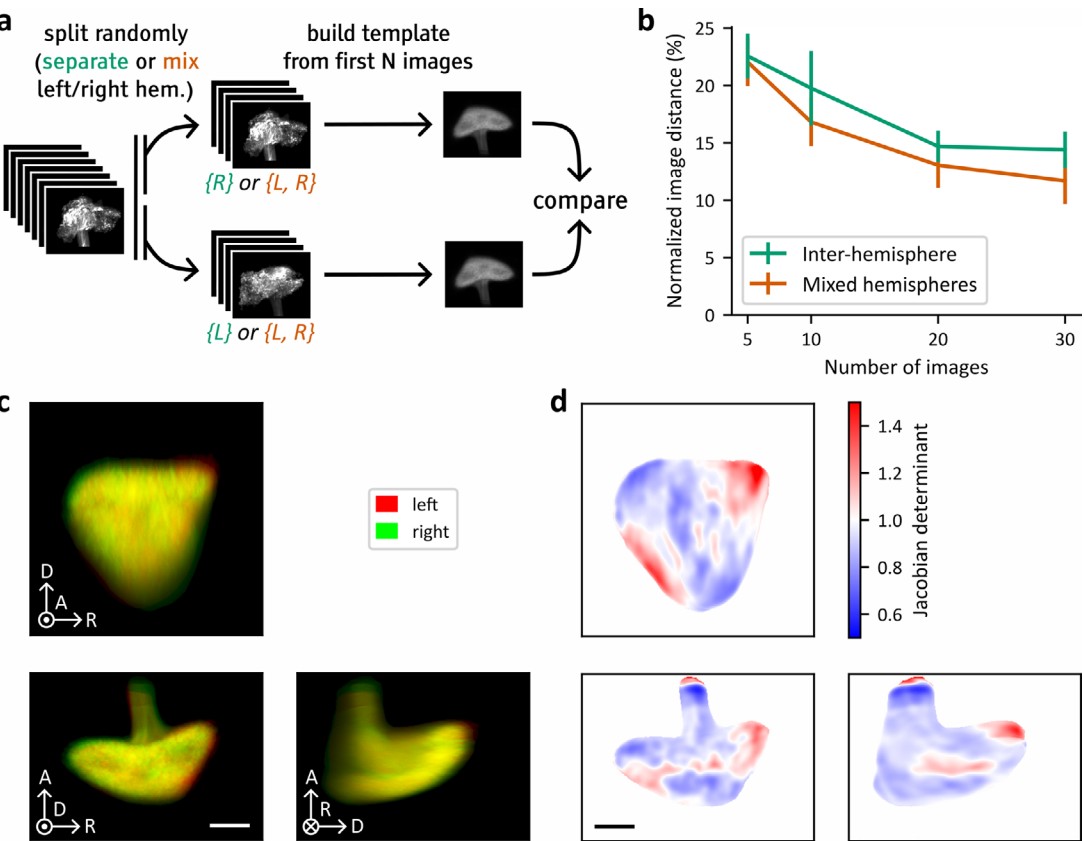

**Appendix 1—figure 2.** Evaluation of intra-subject and intra-hemisphere similarities in the spatial distribution of mushroom body (MB) neuronal subtypes. (**a**) Schematic illustrating the method used to assess differences between templates generated from single or mixed hemispheres. This process used images preprocessed for template building (*Figure 2a*), downsampled by a factor of 2 (using linear interpolation and anti-aliasing filtering) to reduce the computational cost. Step 1: The complete set of preprocessed images is randomly divided into two non-overlapping half sets, arranged in random order. In the mixed hemispheres case, the half sets are created by including both left and right images indiscriminately. In the inter-hemispheres case, the first and second halves exclusively contain left and right MB images, respectively. Step 2: A template is constructed using the first N images from each half set, where N represents the subset size. Step 3: The two resulting templates are compared by registering one to the other using rigid registration and computing the normalized image distance between the aligned templates. This procedure was repeated with eight different random splits performed for each case. (**b**) Comparison of the normalized image distances between templates constructed from subsets of varying sizes, consisting of either mixed or separate hemispheres. In the 30-image case, the mixed-hemisphere distances (n=8, $\mu=11.7\pm2.0\%$) were significantly lower than inter-hemisphere distances (n=8, $\mu=14.4\pm1.6\%$; Student's *t*-test, $t_{14}=2.8$, $p=1.4\times10^{-2}$). (**c**) Overlapped maximum intensity projections of two templates obtained using all available left MB images (in red) and right MB images (in green), respectively. The right MB template was aligned to the left one through rigid registration. Axes indicate anterior (A), dorsal (D) and right (R) anatomical orientations. View-aligned axes orientations depicted with dot (towards viewer) and cross (away from viewer). 25 μm scale bar. (**d**) Mean projections of the determinant of the Jacobian determined from the displacement field obtained through the registration of the right template onto the left template. Values superior to 1 indicate that the left template should be expanded to match the right one. Values inferior to 1 indicate that it should be compressed. Voxels located outside the segmented left MB image were ignored in the average. Pixels representing the mean value of ignored voxels only were set to 1.

## The anticorrelation between $f_{free}$ and $\tau_{bound}$ is not artificially introduced by the fitting process

It has been known long that estimations of parameters of multi-exponential models tend to be correlated, due to the fact that multiple parameters can affect the simulated curve in a similar way (*Grinvald and Steinberg, 1974*; *Johnson, 2000*). In the case of NAD(P)H FLIM, increasing $f_{free}$ or

reducing $\tau_{bound}$ affects the overall fluorescence decay similarly. We hypothesized that this artifact of the fitting process could partially explain why $\tau_{bound}$ and $f_{free}$ values consistently exhibit anticorrelated patterns in multiple previous studies (**Bird et al., 2005**; **Skala et al., 2007**; **Wetzker and Reinhardt, 2019**; **Yang et al., 2021**) and in our data (**Appendix 1—figure 3**). To verify this hypothesis, we simulated 5,000 decay curves using our bi-exponential model, with $\tau_{free}$ set to 0.4 ns. $b$ was fixed to $1.5\times10^{-3}$ to match the overall average of Dataset 2. $\tau_{bound}$ was uniformly sampled between 3 and 4.5 ns and $f_{free}$ between 0.6 and 0.8, similarly to the ranges of within-subject averages obtained from Dataset 2. All simulations used the same arbitrarily selected IRF from Dataset 2 and $\Delta t$ was set to 0. The decay curves were scaled to simulate a photon count of 6000, matching the overall average of Dataset 2. Finally, Poisson-distributed noise was applied. All simulated decay curves were fitted using the same bi-exponential model as in other analyses, with $\tau_{bound}$ and $f_{free}$ as the only parameters. We observed a very low correlation of 0.06 between the estimated values of $\tau_{bound}$ and $f_{free}$ (**Appendix 1—figure 4a**), implying that the anticorrelations observed in our study were unlikely introduced by the fitting process.

As a supplementary verification, we simulated 5000 decay curves from correlated $\tau_{bound}$ and $f_{free}$ couples. Correlated random variables $C_1$ and $C_2$ were obtained from independent random uniform variables $X_1$ and $X_2$ using the Cholesky decomposition:

$$C_1 = X_1$$

$$C_2 = \rho X_1 + \sqrt{1 - \rho^2 X_2}$$

with the Pearson correlation $\rho$ set to –0.86 to match the value obtained from Dataset 2. $C_1$ and $C_2$ were then scaled and offset to vary in the same ranges as $\tau_{bound}$ and $f_{free}$ in the previous simulation. In this simulation, the estimated parameters showed an important negative correlation of –0.64 (**Appendix 1—figure 4b**), confirming the capacity of the fitting process to retrieve dependencies between $\tau_{bound}$ and $f_{free}$.

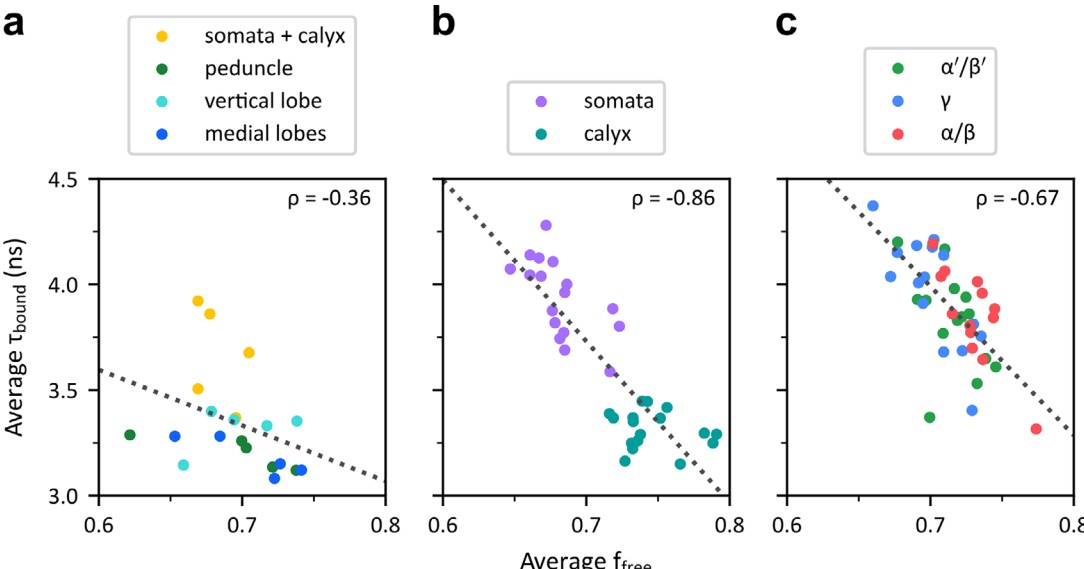

**Appendix 1—figure 3.** Relationship between $f_{free}$ and $\tau_{bound}$ in different datasets. Each dot represents the average values of $f_{free}$ and $\tau_{bound}$ for a single subject. $\rho$ indicates the Pearson correlation coefficient. The dotted lines indicate least squares affine fits. (**a**) Relationship between average $f_{free}$ and $\tau_{bound}$ values of segmented mushroom body (MB) regions, for all subjects in Dataset 1. (**b**) Relationship between average $f_{free}$ and $\tau_{bound}$ values of the somata region, for all subjects in Dataset 2. (**c**) Relationship between average $f_{free}$ and $\tau_{bound}$ values in the different neuronal subtypes, for all subjects in Dataset 6.

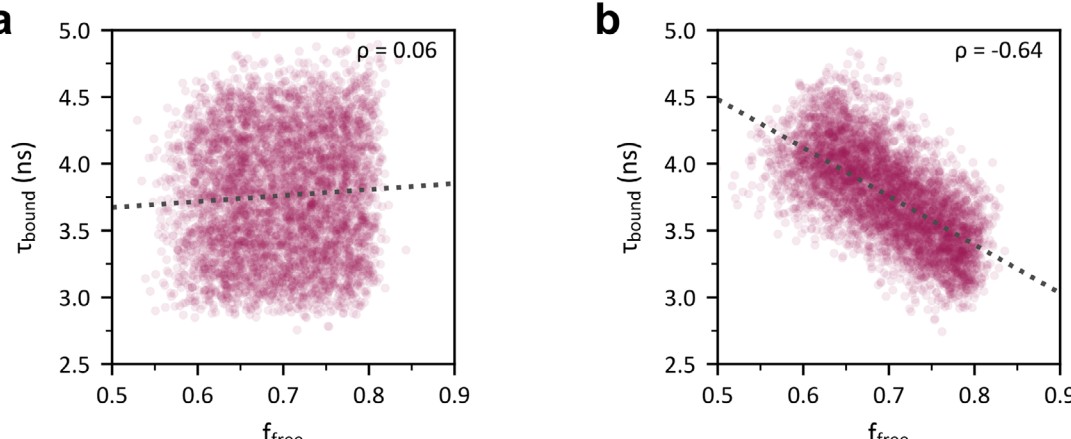

**Appendix 1—figure 4.** Estimations of $\tau_{bound}$ and $f_{free}$ on simulated decays. Each dot represents the values of $f_{free}$ and $\tau_{bound}$ estimated from a single simulated decay. $\rho$ indicates the Pearson correlation coefficient. The dotted line indicates the least squares affine fit. (**a**) Estimations from 5,000 decays simulated by uniformly sampling $f_{free}$ between 0.65 and 0.75 and $\tau_{bound}$ between 3 and 4.5 ns, independently. (**b**) Estimations from 5000 decays simulated by sampling correlated $f_{free}$ and $\tau_{bound}$ couples on the same ranges, with a defined correlation of –0.86.

## Somata of KC subtypes have specific spatial distributions

We computed the average spatial distribution of each of the three KC subtypes within the somata region of the template image (*Appendix 1—figure 5a*). We tested whether the average spatial distribution of the somata of each KC subtype was specific, using images of Dataset 3 featuring a KC-specific marker channel and a subtype-specific marker channel. Image subsets of increasing size were used to compare the average spatial distributions of the different subtypes (*Appendix 1—figure 5b*; Supplementary methods). Average distributions from images of the same subtype consistently showed greater similarity than to those from different subtypes (*Appendix 1—figure 5c*). When using 20 images per spatial distribution, the average differences were 8.1%, 13.5%, and 13.1% for intra-α/β, intra-α'/β', and intra-γ comparisons, respectively, and 19.4% for inter-type comparisons. This result demonstrates that the spatial distribution of KC somata is subtype-specific.

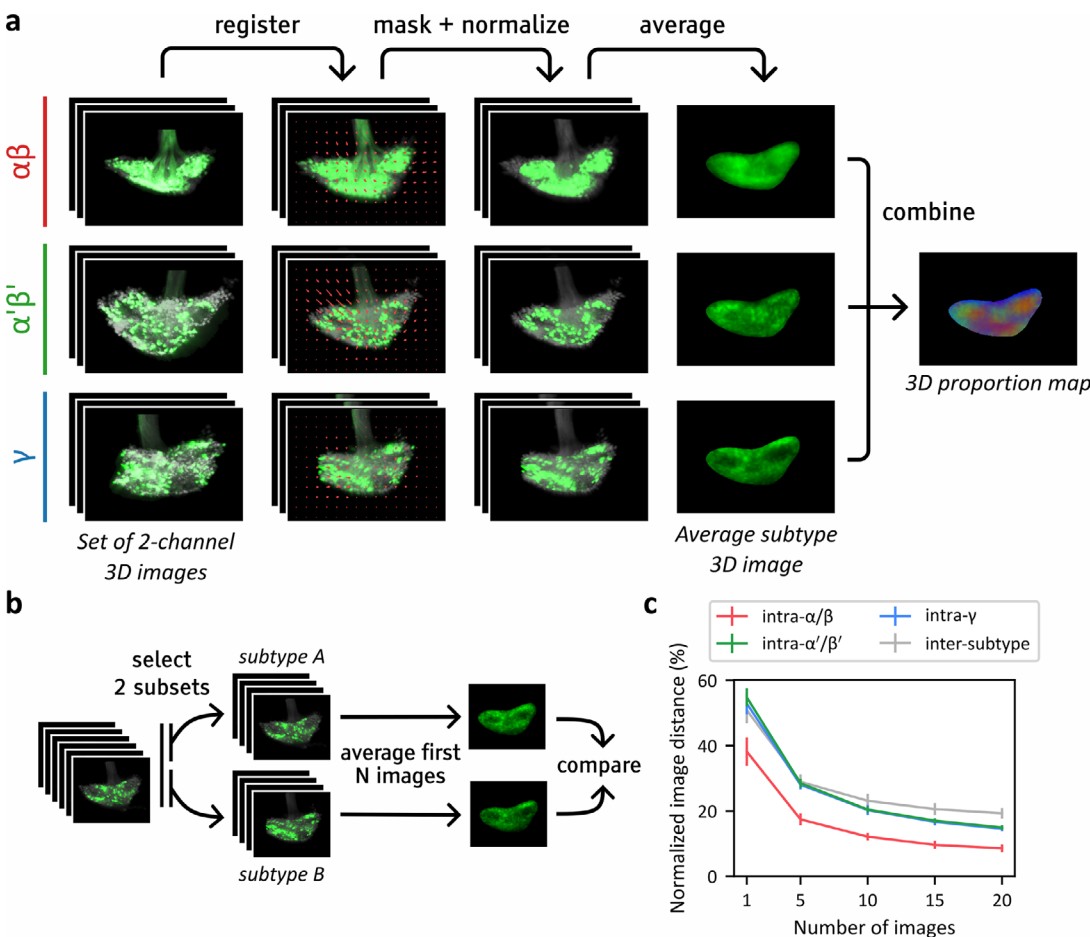

**Appendix 1—figure 5.** Specificity of average spatial distribution of Kenyon cell (KC) subtypes. (**a**) Main steps of the computation of the average spatial distributions of the three main KC subtypes using Dataset 3. Each row shows the processing of a set of images of flies expressing a KC-specific marker (displayed in gray) and a subtype-specific nuclear marker (displayed in green). (**b**) Schematic illustrating the method used to assess differences between average spatial distributions of the different subtypes. (**c**) Comparison of the normalized image distance between average spatial distributions of identical or different subtypes, computed from different numbers of images. The graph reports mean and standard deviations. The inter-subtype distances were compared with the intra-subtype distances for the averages obtained from 20 images. The inter-subtype distances (n=60, μ=19.3±1.6%) were significantly higher than the intra-α/β (n=20, μ=8.6±1.2%; Student's *t*-test, $t_{78}$=26.8, Bonferroni-adjusted *p*=1.3×$10^{-40}$), the intra-α'/β' (n=20, μ=14.9±0.7%; Student's *t*-test, $t_{78}$=11.5, Bonferroni-adjusted *p*=5.9·$10^{-18}$), and the intra-γ distances (n=20, μ=14.6±0.7%; Student's *t*-test, $t_{78}$=12.4, Bonferroni-adjusted *p*=1.0×$10^{-19}$).

## The spatial distributions of KC subtype somata are not hemisphere-specific

We compared the spatial distributions of the somata of KC subtypes in images of Dataset 3 (Supplementary methods). Using pairs of MB images coming from different subjects, the distances between left and right hemispheres were not significantly different from the intra-hemisphere differences (*Appendix 1—figure 6*). This result indicated that the subtype distributions are not hemisphere-specific. This analysis also showed that distances between left and right α/β distributions were lower when they came from the same subject.

**Appendix 1—figure 6.** Measure of intra-subject and intra-hemisphere similarities of the spatial distribution of the mushroom body (MB) neuronal subtypes. Left: Illustration of the different types of image-to-image comparisons on the MBs of two subjects. Right: Image-to-image distances based on 165 MB images of Dataset 3. The intra-subject distances were measured between all pairs of left and right MBs belonging to the same subject (n=48, n=36, and n=42 values for α/β, α'/β', and γ, respectively). The inter-subject/intra-hemisphere distances were measured between all pairs of images of the same hemisphere belonging to different individuals (n=813, n=609, and n=790 values). The inter-subject/inter-hemisphere distances were measured between all pairs of images of different hemispheres belonging to different individuals (n=816, n=598, n=785 values). Statistical tests were performed in order to assess possible intra-subject or intra-hemisphere similarities. It revealed that α/β spatial distributions between left and right hemisphere images are more similar if they belong to the same subject than if they belong to two different subjects (Student's $t$-test, Bonferroni-adjusted $p$=1.4×10$^{-3}$, $t_{838}$=−3.4). No other statistically significant effect was observed.

## The average spatial distributions of KC subtypes are highly overlapping

We have shown that the average spatial distributions of the somata of KC subtypes were different. These distributions can be visualized through the subtype proportion map (*Appendix 1—figure 7a*). As can be observed from the color scale of this map, the proportions of each subtype vary within specific ranges: 20–70%, 10–40%, and 10–60% for α/β, α'/β', and γ, respectively. When looking at the distributions of the proportions, we observe that in most voxels, the proportion of a given subtype corresponds to the overall fraction of neurons belonging to this subtype (*Appendix 1—figure 7b*). This shows that although the distributions are non-uniform, the high inter-subject variability prevents the delineation of areas that consistently isolate specific neuronal subtypes across individual flies.

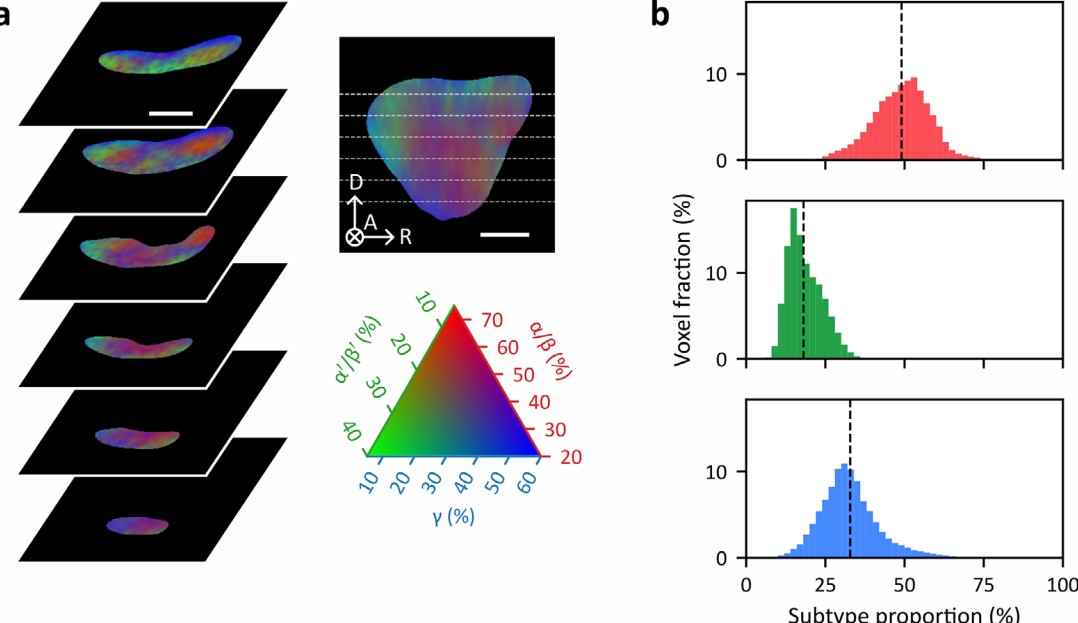

**Appendix 1—figure 7.** Spatial distribution of Kenyon cell (KC) subtype proportions. (**a**) Horizontal slices (left) and posterior projection (upper right) of the subtype proportion map. The proportions of red, green, and blue indicate the proportion of α/β, α'/β', and γ neurons, respectively. The correspondence between color and proportions is indicated by the ternary diagram (lower right). Axes indicate anterior (A), dorsal (D), and right (R) anatomical orientations. The circled cross indicates the direction point away from the viewer. 25 μm scale bar. (**b**) Distribution of the average proportions of each subtype over all voxels. The dashed lines indicate the overall proportion of the subtypes, given by the number of neurons of the considered subtype against the total number of KCs, using the average neuron counts reported by *Aso et al., 2009*.

## The spatial distribution of $f_{free}$ in the MB cortical region is not hemisphere-specific

We compared the spatial distributions of $f_{free}$ in images of Dataset 2 (Supplementary methods). Using pairs of MB images coming from different subjects, the distances between left and right hemispheres were not significantly different from the intra-hemisphere differences (*Appendix 1—figure 8*). This result indicated that the subtype distributions are not hemisphere-specific.

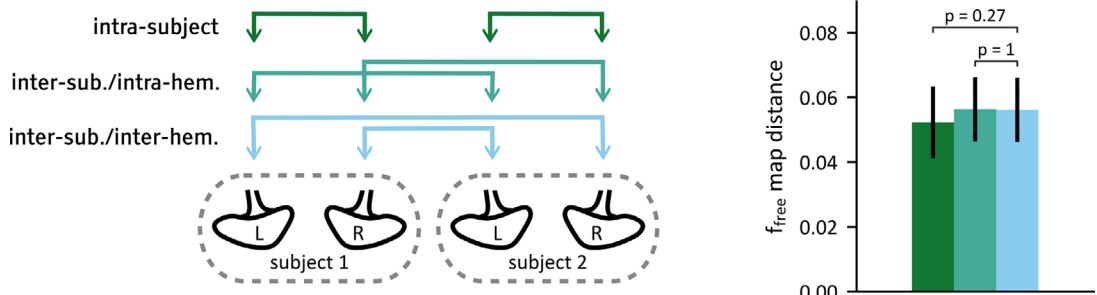

**Appendix 1—figure 8.** Lateralization of $f_{free}$ spatial distribution in the somata area. Left: Illustration of the different types of image-to-image comparisons on the mushroom bodies (MBs) of two subjects. Right: Comparison of single-hemisphere maps of $f_{free}$ in the somata region. The $f_{free}$ map distance was computed between all pairs of registered maps. We compared inter-subject/inter-hemisphere differences (n=224, μ=0.056±0.009) with inter-subject/intra-hemisphere differences (n=227, μ=0.056±0.009) and intra-subject differences (n=14, μ=0.052±0.010). We found that local differences were not significantly different when comparing the same or different hemispheres in different subjects (Student's *t*-test, $t_{449}$=0.13, Bonferroni-adjusted *p*=1.0), suggesting the absence of hemisphere-specific spatial distribution of $f_{free}$ values. We found that local differences were not significantly different either
*Appendix 1—figure 8 continued on next page*

*Appendix 1—figure 8 continued*

when comparing different hemispheres in the same or different subjects (Student's *t*-test, $t_{236}$=1.51, Bonferroni-adjusted $p$=0.27), suggesting that subject-specific spatial distribution of $f_{free}$ are not shared between both hemispheres.

## The mean $f_{free}$ value is a relevant indicator of the shift induced by memory formation in α/β somata

We carried out analyses to verify whether working with mean $f_{free}$ values was relevant or if more complex characteristics of the $f_{free}$ distribution shapes should be taken into account to quantify the effect of memory formation in α/β somata. We considered directly comparing $f_{free}$ distributions using statistical distances, such as Wasserstein metric or Kullback-Leibler divergence. Doing so using appropriate controls could indeed assess the presence of differences in distribution shapes, but these differences would not necessarily be as straightforward to interpret as a shift of the distribution mean. To facilitate interpretation, we decided to employ a parametric approach. The $f_{free}$ distributions of single-hemisphere images across voxels being unimodal with asymmetrical tails (*Appendix 1—figure 9*), we characterized their shapes by extracting the first three statistical moments: mean, variance, and skewness. Around α/β somata, the within-hemisphere means of $f_{free}$ were significantly lower for flies exposed to paired training than for controls, consistent with the previous result obtained on within-subject $f_{free}$ averages (*Appendix 1—figure 10*). Within-hemisphere variances did not appear to differ between the two conditions. Within-hemisphere skews were negative, indicating that the distributions of $f_{free}$ were left-tailed. A memory-induced shift in $f_{free}$ values affecting only a subpopulation of α/β somata should result in differences in skewness between the two conditions. However, no statistically significant difference in skewness was observed.

For ALAT-KD flies, no statistically significant learning-induced difference was observed in any of the first three statistical moments (*Appendix 1—figure 11*).

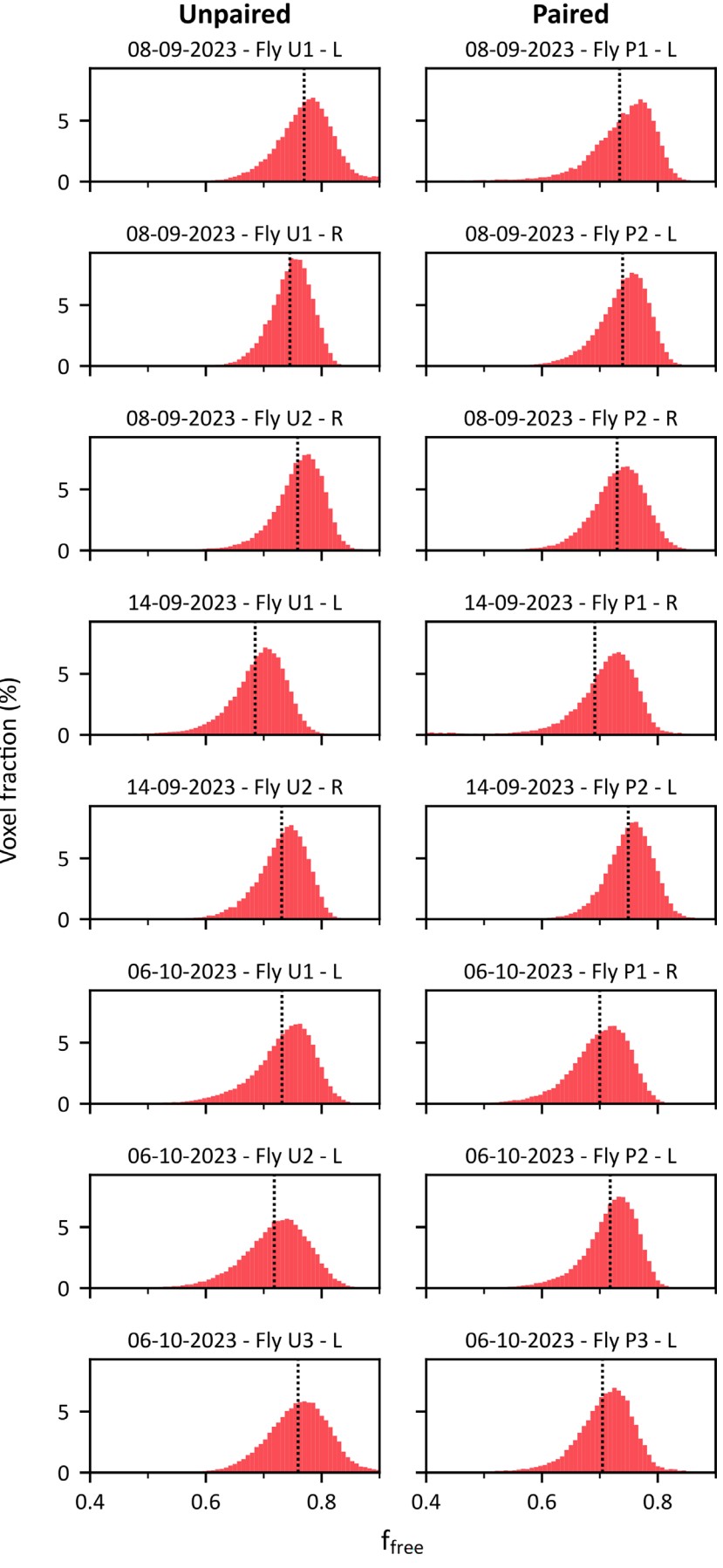

**Appendix 1—figure 9.** Examples of distributions of $f_{free}$ around the somata of α/β neurons. Distributions of $f_{free}$ for single-hemisphere images of Dataset 8, from flies subjected to unpaired and paired conditioning are shown in the left and right columns, respectively. The dotted line indicates the mean value of each distribution.

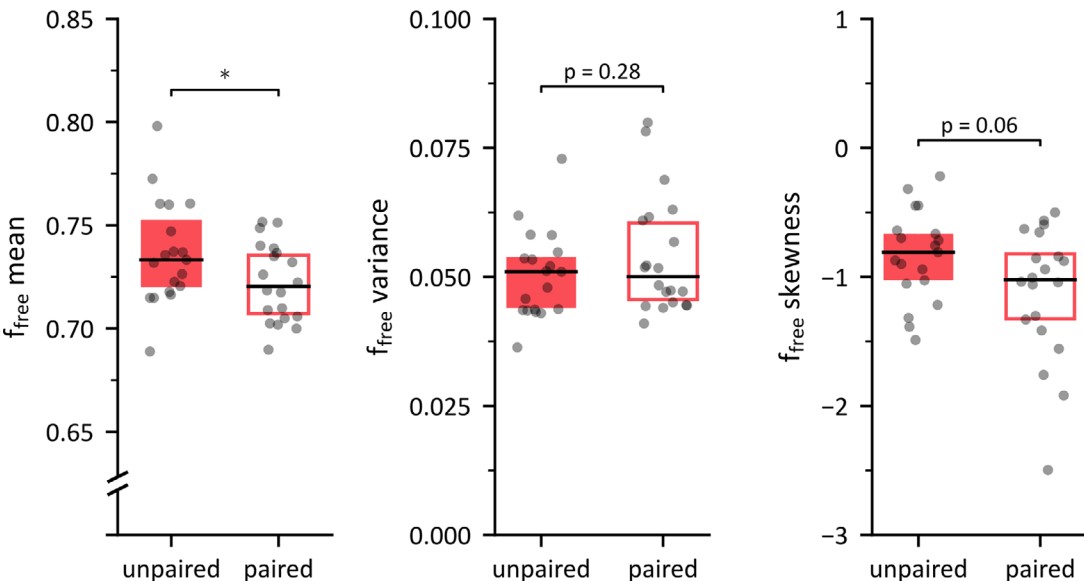

**Appendix 1—figure 10.** Three first moments of the within-hemisphere distributions of $f_{free}$ around the somata of α/β neurons. The mean, variance, and skewness of the distributions of $f_{free}$ were computed for all single-hemisphere images in Dataset 8. The $f_{free}$ mean was significantly higher in the unpaired condition (n=19, μ=0.737±0.025) compared to the paired one (n=20, μ=0.722±0.019; Student's $t$-test, $t_{37}$=2.04, p=4.85×10⁻²). The $f_{free}$ variance was comparable in the unpaired condition (μ=0.050±0.008) and in the paired one (μ=0.054±0.011; Student's $t$-test, $t_{37}$=1.1, p=0.28). The skewness was not significantly different in the unpaired condition (μ=−0.84±0.35) and in the paired one (μ=−1.12±0.51; Student's $t$-test, $t_{37}$=1.98, p=5.54×10⁻²).

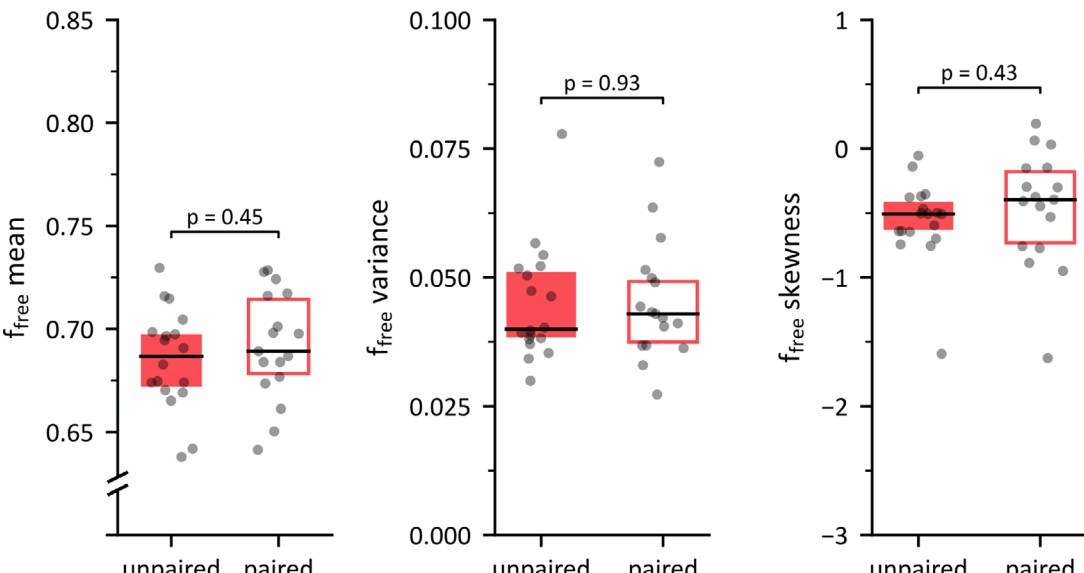

**Appendix 1—figure 11.** Three first moments of the within-hemisphere distributions of $f_{free}$ around the somata of α/β neurons for ALAT-KD flies. The mean, variance, and skewness of the distributions of $f_{free}$ were computed for all single-hemisphere images in Dataset 10. The $f_{free}$ mean was significantly higher in the unpaired condition (n=18, μ=0.685±0.024) compared to the paired one (n=17, μ=0.692±0.026; Student's $t$-test, $t_{33}$=0.76, p=0.45). The $f_{free}$ variance was comparable in the unpaired condition (μ=0.045±0.011) and in the paired one (μ=0.045±0.011; Student's $t$-test, $t_{33}$=0.09, p=0.93). The skewness was not significantly different in the unpaired condition (μ=−0.56±0.32) and in the paired one (μ=−0.46±0.46; Student's $t$-test, $t_{33}$=0.43, p=0.43).

### Estimation of $\tau_{free}$ across the different datasets

$\tau_{free}$ was estimated from FLIM images of the different datasets, using the same preprocessing and masks as in the main results. These estimations were obtained by applying the same fitting procedure as in the other analyses, without fixing $\tau_{free}$. The results are displayed in *Appendix 1—figure 12*.

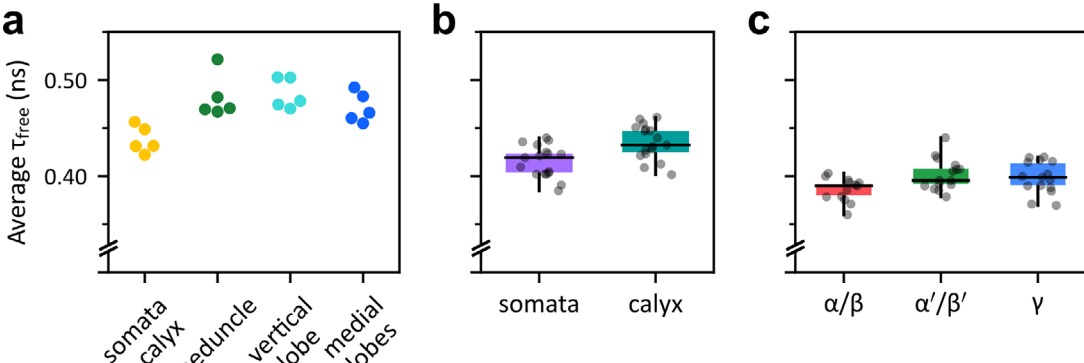

**Appendix 1—figure 12.** Average values of $\tau_{free}$ across the different datasets. Each value represents the within-subject average. (**a**) Average $\tau_{free}$ values in the somata and calyx region (μ=0.438±0.016 ns), the peduncle (μ=0.482±0.023 ns), the vertical lobe (μ=0.485±0.016 ns), and the medial lobes (μ=0.471±0.014), for all subjects in Dataset 1 (n=5). (**b**) Average $\tau_{free}$ values in somata region (μ=0.415±0.017 ns) and in the calyx (μ=0.435±0.018 ns), for all subjects in Dataset 2 (n=17). (**c**) Average $\tau_{bound}$ values near the somata of α/β (n=13, μ=0.386±0.012 ns), α'/β' (n=15, μ=0.0402±0.016 ns), and γ neurons (n=15, μ=0.398±0.017 ns), for all subjects in Dataset 6.

## Supplementary methods

### Image distance metrics

For fluorescence intensity images, we employed a metric termed the 'normalized image distance.' The compared images were first normalized so that their voxel intensities summed to one. The normalized distance was then calculated as the sum of absolute voxel-wise differences between the normalized images, divided by 2. It can be understood as the fraction of the image intensity that should be displaced to match one image to the other. For images A and B, noted $I_A$ and $I_B$, and their summed intensity $S_A$ and $S_B$, this distance can be formulated as:

$$D_{norm}\left(I_A, I_B\right) = \frac{1}{2N_{image}} \sum_{(x,y,z) \in image} \left| \frac{I_A\left(x,y,z\right)}{S_A} - \frac{I_B\left(x,y,z\right)}{S_B} \right|$$

where $N_{image}$ is the number of voxels in the images.

To compare pairs of $f_{free}$ spatial distributions registered in the template space, we used a metric termed the '$f_{free}$ map distance.' This metric was defined as the average of absolute voxel-wise differences across all valid voxels in a pair of $f_{free}$ maps. For the $f_{free}$ maps of images A and B, noted $F_A$ and $F_B$, this distance can be formulated as:

$$D\left(F_A, F_B\right) = \frac{1}{N_{mask}} \sum_{(x,y,z) \in mask} \left| F_A\left(x,y,z\right) - F_B\left(x,y,z\right) \right|$$

where $N_{mask}$ is the number of voxels in the mask.

### Building a map of subtype proportions

A map, estimating the average proportion of each KC subtype over the MB template was derived from Dataset 3. The images of this set are composed of 2 channels, one labeling all KCs with a cytosolic marker and the other labeling KCs from one KC subtype with a nuclear marker. Right-hemisphere images were mirrored. All images were registered to the template based on the KC-specific marker channel. We applied a previously established mask (*Figure 2—figure supplement 1*) to isolate the signal of the somata region in the subtype-specific channel. The masked images

were normalized to make the total intensity of each image equal to the number of neurons of the associated subtypes (1002 α/β, 370 α'/β', and 671 γ neurons on average, according to *Aso et al., 2009*). The intensity of the normalized images can be seen as an approximation of the neuronal density, expressed in neuron/voxel. The three subtype density maps are then summed to produce a KC density map. Finally, each subtype density map is divided voxel-wise by the KC density map to produce a subtype proportion map.

## Comparing hemisphere-specific MB shapes

This process was carried out using images of Dataset 3. All images were preprocessed following the same steps as for template building (*Figure 2a*). Our testing process involved repeated distance measurements (*Appendix 1—figure 2a*). The complete set of preprocessed images was first randomly divided into two non-overlapping subsets, arranged in random order. In the mixed-hemisphere case, the half sets were created from left and right images indiscriminately. In the inter-hemisphere case, the first and second subset exclusively contained left and right MB images, respectively. A template was constructed using the first N images from each subset. The resulting average spatial distributions were compared by computing the total absolute difference between them. This procedure was repeated multiple times, with five random splits performed for each case.

## Testing the hemisphere-specificity of the spatial distributions of KC subtype somata

This process was carried out using images of Dataset 3. All images were registered and masked following the same steps as for the establishment of the subtype proportion map (*Appendix 1—figure 10a*). The normalized image distance was computed between all pairs of images. All distances were labeled as either intra-subject, inter-subject, intra-hemisphere, or inter-subject, inter-hemisphere.

## Testing the specificity of the spatial distributions of KC subtype somata

This process was carried out using images of Dataset 3. All images were registered, masked, and normalized following the same steps as for the establishment of the subtype proportion map (*Appendix 1—figure 5a*). Our testing process involved repeated distance measurements (*Appendix 1—figure 5b*). For each measurement, the dataset was split into two non-overlapping subsets of images. Each set was composed of images displaying a unique subtype-specific marker. Intra-type comparisons were computed using two sets of the same subtype, while extra-type comparisons used sets of different subtypes. In both cases, the first N images of each set were averaged. Finally, the total absolute difference was computed between the two resulting average spatial distributions. This scheme was repeated multiple times for different random splits and values of N. For each neuronal subtypes, 20 splits were used.

## Testing the hemisphere-specificity of the spatial distribution of $f_{free}$ in the somata region

This process was carried out using images of Dataset 2. Right-hemisphere images were mirrored. Maps of $f_{free}$ in the somata region were obtained by registering the corresponding mask to each image. The resulting $f_{free}$ maps were registered to the template using the KC-specific marker channel. The total absolute difference was computed between all pairs of registered $f_{free}$ maps. All distances were labeled as intra-subject, inter-subject, intra-hemisphere, or inter-subject inter-hemisphere (*Appendix 1—figure 8*).

## Appendix 2

**Appendix 2—table 1.** Composition of dataset 2.

|  | Subjects | Left images | Right images | Total images |
|---|---|---|---|---|
| Total | 17 | 14 | 17 | 31 |

**Appendix 2—table 2.** Composition of dataset 3.

|  | Subjects | Left images | Right images | Total images |
|---|---|---|---|---|
| α/β | 34 | 28 | 30 | 58 |
| α'/β' | 32 | 22 | 28 | 50 |
| γ | 36 | 26 | 31 | 57 |
| Total | 102 | 76 | 89 | 165 |

**Appendix 2—table 3.** Composition of dataset 4.

|  | Subjects | Left images | Right images | Total images |
|---|---|---|---|---|
| Total | 19 | 16 | 16 | 32 |

**Appendix 2—table 4.** Composition of dataset 5.

|  | Subjects | Left images | Right images | Total images |
|---|---|---|---|---|
| Unpaired | 15 | 8 | 7 | 15 |
| Paired | 16 | 8 | 8 | 16 |
| Total | 31 | 16 | 15 | 31 |

**Appendix 2—table 5.** Composition of dataset 6.

|  | Subjects | Left images | Right images | Total images |
|---|---|---|---|---|
| α/β | 13 | 13 | 12 | 25 |
| α'/β' | 15 | 14 | 14 | 28 |
| γ | 15 | 12 | 14 | 26 |
| Total | 43 | 39 | 40 | 79 |

**Appendix 2—table 6.** Composition of dataset 7.

|  | Subjects | Left images | Right images | Total images |
|---|---|---|---|---|
| Control | 12 | 6 | 6 | 12 |
| Ldh-knockdown | 12 | 7 | 5 | 12 |
| Total | 24 | 13 | 11 | 24 |

**Appendix 2—table 7.** Composition of dataset 8.

|  | Subjects | Left images | Right images | Total images |
|---|---|---|---|---|
| Unpaired | 18 | 10 | 9 | 19 |
| Paired | 19 | 10 | 10 | 20 |
| Total | 37 | 20 | 19 | 39 |

*Appendix 2—table 8 Continued on next page*

**Appendix 2—table 8.** Composition of dataset 9.

|  | Subjects | Left images | Right images | Total images |
|---|---|---|---|---|
| Unpaired | 11 | 6 | 5 | 11 |
| Paired | 12 | 5 | 7 | 12 |
| Total | 23 | 11 | 12 | 23 |

**Appendix 2—table 9.** Composition of dataset 10.

|  | Subjects | Left images | Right images | Total images |
|---|---|---|---|---|
| Unpaired | 18 | 8 | 10 | 18 |
| Paired | 17 | 8 | 9 | 17 |
| Total | 35 | 14 | 19 | 35 |

