## [Editor Report · eLife Assessment]

This **valuable** study uses NAD(P)H fluorescence lifetime imaging (FLIM) to map metabolic states in the Drosophila brain. The authors reveal subtype-specific metabolic profiles in Kenyon cells and report learning-related changes, supported by **solid** evidence and careful methodology. However, the FLIM shifts observed after memory formation in α/β neurons are small and only weakly significant, so the ability of FLIM to detect subtle physiological changes still requires further validation. Nevertheless, this work provides a strong starting point and demonstrates the promising potential of FLIM for probing neural metabolism in vivo.

---

## [Referee Report · Reviewer #1 (Public review)]

Summary:

The authors present a novel usage of fluorescence life-time imaging microscopy (FLIM) to measure NAD(P)H autofluorescence in the Drosophila brain, as a proxy for cellular metabolic/redox states. This new method relies on the fact that both NADH and NADPH are autofluorescent, with a different excitation lifetime depending on whether they are free (indicating glycolysis) or protein-bound (indicating oxidative phosphorylation). The authors successfully use this method in Drosophila to measure changes in metabolic activity across different areas of the fly brain, with a particular focus on the main center for associative memory: the mushroom body.

Strengths:

The authors have made a commendable effort to explain the technical aspects of the method in accessible language. This clarity will benefit both non-experts seeking to understand the methodology and researchers interested in applying FLIM to Drosophila in other contexts.

Weaknesses:

Despite being statistically significant, the learning-induced change in f-free in α/β Kenyon cells is minimal (a decrease from 0.76 to 0.73, with a high variability). It is unclear whether this small effect represents a meaningful shift in neuronal metabolic state.

Whether this method can be valuable to examine the effects of long-term memory (after spaced or massed conditioning) remains to be established.

---

## [Referee Report · Reviewer #2 (Public review)]

This revised manuscript presents a valuable application of NAD(P)H fluorescence lifetime imaging (FLIM) to study metabolic activity in the Drosophila brain. The authors reveal regional differences in oxidative and glycolytic metabolism, with particular emphasis on the mushroom body, a key center for associative learning and memory. They also report metabolic shifts in α/β Kenyon cells following classical conditioning, in line with their known role in energy-demanding memory processes.

The study is well-executed and the authors have added more detailed methodological descriptions in this version, which strengthen the technical contribution. The analysis pipeline is rigorous, with careful curve fitting and appropriate controls. However, the metabolic shifts observed after conditioning are small and only weakly significant, raising questions about the sensitivity of FLIM for detecting subtle physiological changes. The authors acknowledge these limitations in the revised discussion, which helps place the findings in proper context.

Despite this, the work provides a solid foundation for future applications of label-free FLIM in vivo and serves as a valuable technical resource for researchers interested in neural metabolism. Overall, this study represents a meaningful step toward integrating metabolic imaging with the study of neural activity and cognitive function.

---

## [Referee Report · Reviewer #3 (Public review)]

This study investigates the characteristics of the autofluorescence signal excited by 740 nm 2-photon excitation, in the range of 420-500 nm, across the Drosophila brain. The fluorescence lifetime (FL) appears bi-exponential, with a short 0.4 ns time constant followed by a longer decay. The lifetime decay and the resulting parameter fits vary across the brain. The resulting maps reveal anatomical landmarks, which simultaneous imaging of genetically encoded fluorescent proteins help identify. Past work has shown that the autofluorescence decay time course reflects the balance of the redox enzyme NAD(P)H vs. its protein bound form. The ratio of free to bound NADPH is thought to indicate relative glycolysis vs. oxidative phosphorylation, and thus shifts in the free-to-bound ratio may indicate shifts in metabolic pathways. The basics of this measure have been demonstrated in other organisms, and this study is the first to use the FLIM module of the STELLARIS 8 FALCON microscope from Leica to measure autofluorescence lifetime in the brain of the fly. Methods include registering brains of different flies to a common template and masking out anatomical regions of interest using fluorescence proteins.

The analysis relies on fitting a FL decay model with two free parameters, f_free and T_bound. F_free is the fraction of the normalized curve contributed by a decaying exponential with a time constant 0.4 ns, thought to represent the FL of free NADPH or NADH, which apparently cannot be distinguished. T_bound is the time constant of the second exponential, with scalar amplitude = (1-f_free). The T_bound fit is thought to represent the decay time constant of protein bound NADPH, but can differ depending on the protein. The study shows that across the brain, T_bound can range from 0 to >5 ns, whereas f_free can range from 0.5 to 0.9 ns (Figure 1a). The paper beautifully lays out the analysis pipeline, providing a valuable resource. The full range of fits are reported, including maximum likelihood quality parameters, and can be benchmarks for future studies.

The authors measure properties of NADPH related autofluorescence of Kenyon Cells (KCs) of the fly mushroom body. The somata and calyx of mushroom bodies have a longer average tau_bound than other regions (Figure 1e); the f_free fit is higher for the calyx (input synapses) region than for KC somata; and the average across flies of average f_free fits in alpha/beta KC somata decreases slightly following paired presentation of odor and shock, compared to unpaired presentation of the same stimuli. Though the change is slight, no comparable change is detected in gamma KCs, suggesting that distributions of f_free derived from FL may be sensitive enough to measure changes in metabolic pathways following conditioning.

FLIM as a method is not yet widely prevalent in fly neuroscience, but recent demonstrations of its potential are likely to increase its use. Future efforts will benefit from the description of the properties of the autofluorescence signal to evaluate how autofluorescence may impact measures of FL of genetically engineered indicators.

---

## [Author Response]

The following is the authors’ response to the original reviews.

**Public Reviews:**

**Reviewer #1 (Public review):**
Summary:The authors present a novel usage of fluorescence lifetime imaging microscopy (FLIM) to measure NAD(P)H autofluorescence in the Drosophila brain, as a proxy for cellular metabolic/redox states. This new method relies on the fact that both NADH and NADPH are autofluorescent, with a different excitation lifetime depending on whether they are free (indicating glycolysis) or protein-bound (indicating oxidative phosphorylation). The authors successfully use this method in Drosophila to measure changes in metabolic activity across different areas of the fly brain, with a particular focus on the main center for associative memory: the mushroom body.Strengths:The authors have made a commendable effort to explain the technical aspects of the method in accessible language. This clarity will benefit both non-experts seeking to understand the methodology and researchers interested in applying FLIM to Drosophila in other contexts.Weaknesses:(1) Despite being statistically significant, the learning-induced change in f-free in α/β Kenyon cells is minimal (a decrease from 0.76 to 0.73, with a high variability). The authors should provide justification for why they believe this small effect represents a meaningful shift in neuronal metabolic state.

We agree with the reviewer that the observed f_free shift averaged per individual, while statistically significant, is small. However, to our knowledge, this is the first study to investigate a physiological (i.e., not pharmacologically induced) variation in neuronal metabolism using FLIM. As such, there are no established expectations regarding the amplitude of the effect. In the revised manuscript, we have included an additional experiment involving the knockdown of ALAT in α/β Kenyon cells, which further supports our findings. We have also expanded the discussion to expose two potential reasons why this effect may appear modest.

(2) The lack of experiments examining the effects of long-term memory (after spaced or massed conditioning) seems like a missed opportunity. Such experiments could likely reveal more drastic changes in the metabolic profiles of KCs, as a consequence of memory consolidation processes.

We agree with the reviewer that investigating the effects of long-term memory on metabolism represent a valuable future path of investigation. An intrinsic caveat of autofluorescence measurement, however, is to identify the cellular origin of the observed changes. To this respect, long-term memory formation is not an ideal case study as its essential feature is expected to be a metabolic activation localized to Kenyon cells’ axons in the mushroom body vertical lobes (as shown in Comyn et al., 2024), where many different neuron subtypes send intricate processes. This is why we chose to first focus on middle-term memory, where changes at the level of the cell bodies could be expected from our previous work (Rabah et al., 2022). But our pioneer exploration of the applicability of NAD(P)H FLIM to brain metabolism monitoring in vivo now paves the way to extending it to the effect of other forms of memory.

(3) The discussion is mostly just a summary of the findings. It would be useful if the authors could discuss potential future applications of their method and new research questions that it could help address.

The discussion has been expanded by adding interpretations of the findings and remaining challenges.

**Reviewer #2 (Public review):**
This manuscript presents a compelling application of NAD(P)H fluorescence lifetime imaging (FLIM) to study metabolic activity in the Drosophila brain. The authors reveal regional differences in oxidative and glycolytic metabolism, with a particular focus on the mushroom body, a key structure involved in associative learning and memory. In particular, they identify metabolic shifts in α/β Kenyon cells following classical conditioning, consistent with their established role in energy-demanding middle- and long-term memories.These results highlight the potential of label-free FLIM for in-vivo neural circuit studies, providing a powerful complement to genetically encoded sensors. This study is well-conducted and employs rigorous analysis, including careful curve fitting and well-designed controls, to ensure the robustness of its findings. It should serve as a valuable technical reference for researchers interested in using FLIM to study neural metabolism in vivo. Overall, this work represents an important step in the application of FLIM to study the interactions between metabolic processes, neural activity, and cognitive function.
**Reviewer #3 (Public review):**
This study investigates the characteristics of the autofluorescence signal excited by 740 nm 2-photon excitation, in the range of 420-500 nm, across the Drosophila brain. The fluorescence lifetime (FL) appears bi-exponential, with a short 0.4 ns time constant followed by a longer decay. The lifetime decay and the resulting parameter fits vary across the brain. The resulting maps reveal anatomical landmarks, which simultaneous imaging of genetically encoded fluorescent proteins helps to identify. Past work has shown that the autofluorescence decay time course reflects the balance of the redox enzyme NAD(P)H vs. its protein-bound form. The ratio of free-to-bound NADPH is thought to indicate relative glycolysis vs. oxidative phosphorylation, and thus shifts in the free-to-bound ratio may indicate shifts in metabolic pathways. The basics of this measure have been demonstrated in other organisms, and this study is the first to use the FLIM module of the STELLARIS 8 FALCON microscope from Leica to measure autofluorescence lifetime in the brain of the fly. Methods include registering the brains of different flies to a common template and masking out anatomical regions of interest using fluorescence proteins.The analysis relies on fitting an FL decay model with two free parameters, f_free and t_bound. F_free is the fraction of the normalized curve contributed by a decaying exponential with a time constant of 0.4 ns, thought to represent the FL of free NADPH or NADH, which apparently cannot be distinguished. T_bound is the time constant of the second exponential, with scalar amplitude = (1-f_free). The T_bound fit is thought to represent the decay time constant of protein-bound NADPH but can differ depending on the protein. The study shows that across the brain, T_bound can range from 0 to >5 ns, whereas f_free can range from 0.5 to 0.9 (Figure 1a). These methods appear to be solid, the full range of fits are reported, including maximum likelihood quality parameters, and can be benchmarks for future studies.The authors measure the properties of NADPH-related autofluorescence of Kenyon Cells(KCs) of the fly mushroom body. The results from the three main figures are:(1) Somata and calyx of mushroom bodies have a longer average tau_bound than other regions (Figure 1e);(2) The f_free fit is higher for the calyx (input synapses) region than for KC somata (Figure 2b);(3) The average across flies of average f_free fits in alpha/beta KC somata decreases from 0.734 to 0.718. Based on the first two findings, an accurate title would be "Autofluorecense lifetime imaging reveals regional differences in NADPH state in Drosophila mushroom bodies."The third finding is the basis for the title of the paper and the support for this claim is unconvincing. First, the difference in alpha/beta f_free (p-value of 4.98E-2) is small compared to the measured difference in f_free between somas and calyces. It's smaller even than the difference in average soma f_free across datasets (Figure 2b vs c). The metric is also quite derived; first, the model is fit to each (binned) voxel, then the distribution across voxels is averaged and then averaged across flies. If the voxel distributions of f_free are similar to those shown in Supplementary Figure 2, then the actual f_free fits could range between 0.6-0.8. A more convincing statistical test might be to compare the distributions across voxels between alpha/beta vs alpha'/beta' vs. gamma KCs, perhaps with bootstrapping and including appropriate controls for multiple comparisons.

The difference observed is indeed modest relative to the variability of f_free measurements in other contexts. The fact that the difference observed between the somata region and the calyx is larger is not necessarily surprising. Indeed, these areas have different anatomical compositions that may result in different basal metabolic profiles. This is suggested by Figure 1b which shows that the cortex and neuropile have different metabolic signatures. Differences in average f_free values in the somata region can indeed be observed between naive and conditioned flies. However, all comparisons in the article were performed between groups of flies imaged within the same experimental batches, ensuring that external factors were largely controlled for. This absence of control makes it difficult to extract meaningful information from the comparison between naive and conditioned flies.

We agree with the reviewer that the choice of the metric was indeed not well justified in the first manuscript. In the new manuscript, we have tried to illustrate the reasons for this choice with the example of the comparison of f_free in alpha/beta neurons between unpaired and paired conditioning (Dataset 8). First, the idea of averaging across voxels is supported by the fact that the distributions of decay parameters within a single image are predominantly unimodal. Examples for Dataset 8 are now provided in the new Sup. Figure 14. Second, an interpretable comparison between multiple groups of distributions is, to our knowledge, not straightforward to implement. It is now discussed in Supplementary information. To measure interpretable differences in the shapes of the distributions we computed the first three moments of distributions of f_free for Dataset 8 and compared the values obtained between conditions (see Supplementary information and new Sup. Figure 15). Third, averaging across individuals allows to give each experimental subject the same weight in the comparisons.

I recommend the authors address two concerns. First, what degree of fluctuation in autofluorescence decay can we expect over time, e.g. over circadian cycles? That would be helpful in evaluating the magnitude of changes following conditioning. And second, if the authors think that metabolism shifts to OXPHOS over glycolosis, are there further genetic manipulations they could make? They test LDH knockdown in gamma KCs, why not knock it down in alpha/beta neurons? The prediction might be that if it prevents the shift to OXPHOS, the shift in f_free distribution in alpha/beta KCs would be attenuated. The extensive library of genetic reagents is an advantage of working with flies, but it comes with a higher standard for corroborating claims.

In the present study, we used control groups to account for broad fluctuations induced by external factors such as the circadian cycle. We agree with the reviewer that a detailed characterization of circadian variations in the decay parameters would be valuable for assessing the magnitude of conditioning-induced shifts. We have integrated this relevant suggestion in the Discussion. Conducting such an investigation lies unfortunately beyond the scope and means of the current project.

In line with the suggestion of the reviewer, we have included a new experiment to test the influence of the knockdown of ALAT on the conditioning-induced shift measured in alpha/beta neurons. This choice is motivated in the new manuscript. The obtained result shows that no shift is detected in the mutant flies, in accordance with our hypothesis.

FLIM as a method is not yet widely prevalent in fly neuroscience, but recent demonstrations of its potential are likely to increase its use. Future efforts will benefit from the description of the properties of the autofluorescence signal to evaluate how autofluorescence may impact measures of FL of genetically engineered indicators.
**Recommendations for the authors**

**Reviewer #1 (Recommendations for the authors):**
(1) Y axes in Figures 1e, 2c, 3b,c are misleading. They must start at 0.

Although we agree that making the Y axes start at 0 is preferable, in our case it makes it difficult to observe the dispersion of the data at the same time (your next suggestion). To make it clearer to the reader that the axes do not start at 0, a broken Y-axis is now displayed in every concerned figure.

(2) These same plots should have individual data points represented, for increased clarity and transparency.

Individual data points were added on all boxplots.

**Reviewer #2 (Recommendations for the authors):**
I am evaluating this paper as a fly neuroscientist with experience in neurophysiology, including calcium imaging. I have little experience with FLIM but anticipate its use growing as more microscopes and killer apps are developed. From this perspective, I value the opportunity to dig into FLIM and try to understand this autofluorescence signal. I think the effort to show each piece of the analysis pipeline is valuable. The figures are quite beautiful and easy to follow. My main suggestion is to consider moving some of the supplemental data to the main figures. eLife allows unlimited figures, moving key pieces of the pipeline to the main figures would make for smoother reading and emphasize the technical care taken in this study.

We thank the reviewer for their feedback. Following their advice we have moved panels from the supplementary figures to the main text (see new Figure 2).

Unfortunately, the scientific questions and biological data do not rise to the typical standard in the field to support the claims in the title, "In vivo autofluorescence lifetime imaging of the Drosophila brain captures metabolic shifts associated with memory formation". The authors also clearly state what the next steps are: "hypothesis-driven approaches that rely on metabolite-specific sensors" (Intro). The advantage of fly neuroscience is the extensive library of genetic reagents that enable perturbations. The key manipulation in this study is the electric shock conditioning paradigm that subtly shifts the distribution of a parameter fit to an exponential decay in the somas of alpha/beta KCs vs others. This feels like an initial finding that deserves follow-up; but is it a large enough result to motivate a future student to pick this project up? The larger effect appears to be the gradients in f_free across KCs overall (Figure 2b). How does this change with conditioning?

We acknowledge that the observed metabolic shift is modest relative to the variability of f_free and agree that additional corroborating experiments would further strengthen this result. Nevertheless, we believe it remains a valid and valuable finding that will be of interest to researchers in the field. The reviewer is right in pointing out that the gradient across KCs is higher in magnitude, however, the fact that this technique can also report experience-dependent changes, in addition to innate heterogeneities across different cell types, is a major incentive for people who could be interested in applying NAD(P)H FLIM in the future. For this reason, we consider it appropriate to retain mention of the memory-induced shift in the title, while making it less assertive and adding a reference to the structural heterogeneities of f_free revealed in the study. We have also rephrased the abstract to adopt a more cautious tone and expanded the discussion to clarify why a low-magnitude shift in f_free can still carry biological significance in this context. Finally, we have added the results of a new set of data involving the knockdown of ALAT in Kenyon cells, to further support the relevance of our observation relative to memory formation, despite its small magnitude. We believe that these elements together form a good basis for future investigations and that the manuscript merits publication in its present form.

Together, I would recommend reshaping the paper as a methods paper that asks the question, what are the spatial properties of NADPH FL across the brain? The importance of this question is clear in the context of other work on energy metabolism in the MBs. 2P FLIM will likely always have to account for autofluorescence, so this will be of interest. The careful technical work that is the strength of the manuscript could be featured, and whether conditioning shifts f_free could be a curio that might entice future work.

By transferring panels of the supplementary figures to the main text (see new Figure 2) as suggested by Reviewer 2, we have reinforced the methodological part of the manuscript. For the reasons explained above, we however still mention the ‘biological’ findings in the title and abstract.

Minor recommendations on science:Figure 2C. Plotting either individual data points or distributions would be more convincing.

Individual data points were added on all boxplots.

There are a few mentions of glia. What are the authors' expectations for metabolic pathways in glia vs. neurons? Are glia expected to use one more than the other? The work by Rabah suggests it should be different and perhaps complementary to neurons. Can a glial marker be used in addition to KC markers? This seems crucial to being able to distinguish metabolic changes in KC somata from those in glia.

Drosophila cortex glia are thought to play a similar role as astrocytes in vertebrates (see Introduction). In that perspective, we expect cortex glia to display a higher level of glycolysis than neurons. The work by Rabah et al. is coherent with this hypothesis. Reviewer 2 is right in pointing out that using a glial marker would be interesting. However, current technical limitations make such experiments challenging. These limitations are now exposed in the discussion.

The question of whether KC somata positions are stereotyped can probably be answered in other ways as well. For example, the KCs are in the FAFB connectomic data set and the hemibrain. How do the somata positions compare?

The reviewer’s suggestion is indeed interesting. However, the FAFB and hemibrain connectomic datasets are based on only two individual flies, which probably limits their suitability for assessing the stereotypy of KC subtype distributions. In addition, aligning our data with the FAFB dataset would represent substantial additional work.

The free parameter tau_bound is mysterious if it can be influenced by the identity of the protein. Are there candidate NADPH binding partners that have a spatial distribution in confocal images that could explain the difference between somas and calyx?

There are indeed dozens of NADH- or NADPH-binding proteins. For this reason, in all studies implementing exponential fitting of metabolic FLIM data, tau_bound is considered a complex combination of the contributions from many different proteins. In addition, one should keep in mind that the number of cell types contributing to the autofluorescence signal in the mushroom body calyx (Kenyon cells, astrocyte-like and ensheathing glia, APL neurons, olfactory projection neurons, dopamine neurons) is much higher than in the somas (only Kenyon cells and cortex glia). This could also participate in the observed difference. Hence, focusing on intracellular heterogeneities of potential NAD(P)H binding partners seems premature at that stage.

The phrase "noticeable but not statistically significant" is misleading.

We agree with the reviewer and have removed “noticeable but” from the sentence in the new version of the manuscript.

Minor recommendations on presentation:The Introduction can be streamlined.

We agree that some parts of the Introduction can seem a bit long for experts of a particular field. However, we think that this level of detail makes the article easily accessible for neuroscientists working on Drosophila and other animal models but not necessarily with FLIM, as well as for experts in energy metabolism that may be familiar with FLIM but not with Drosophila neuroscience.